# EFFICIENT BEST-OF-BOTH-WORLDS ALGORITHMS FOR CONTEXTUAL COMBINATORIAL SEMI-BANDITS

**Mengmeng Li**
Risk Analytics and Optimization Chair
EPFL
Lausanne, Switzerland
mengmeng.li@epfl.ch

**Philipp J. Schneider**
Risk Analytics and Optimization Chair
EPFL
Lausanne, Switzerland
philipp.schneider@epfl.ch

**Jelisaveta Aleksic**
Risk Analytics and Optimization Chair
EPFL
Lausanne, Switzerland
jelisaveta.aleksic@epfl.ch

**Daniel Kuhn**
Risk Analytics and Optimization Chair
EPFL
Lausanne, Switzerland
daniel.kuhn@epfl.ch

## ABSTRACT

We introduce the first best-of-both-worlds algorithm for contextual combinatorial bandits that simultaneously guarantees $\widetilde{\mathcal{O}}(\sqrt{T})$ regret in the adversarial regime and $\widetilde{\mathcal{O}}(\ln T)$ regret in the corrupted stochastic regime. Our approach builds on the Follow-the-Regularized-Leader (FTRL) framework equipped with a negative Shannon entropy regularizer, yielding a flexible method that admits efficient implementations. Beyond regret bounds, we tackle the practical bottleneck in FTRL (or, equivalently, Online Stochastic Mirror Descent) arising from the high-dimensional projection step encountered in each round of interaction. By leveraging the Karush-Kuhn-Tucker conditions, we transform the $K$-dimensional convex projection problem into a single-variable root-finding problem, dramatically accelerating each round. Empirical evaluations demonstrate that this combined strategy not only attains the attractive regret bounds of best-of-both-worlds algorithms but also delivers substantial per-round speed-ups, making it well-suited for large-scale, real-time applications.

## 1 INTRODUCTION

Online decision-making is often modeled via the *multi-armed bandit* framework: over $T$ rounds, a learner selects an action and incurs a loss, observing only partial feedback. Many real-world tasks—from selecting at most $m$ movies on a streaming platform to curating a list of $m$ products on an e-commerce homepage—require choosing a subset of up to $m$ items from $K \gg m$ base arms in each round. This *combinatorial bandit* variant admits either *semi-bandit feedback* (per-arm losses) or *full-bandit feedback* (aggregate loss). Semi-bandit feedback is realistic—web analytics record click-through outcomes for each displayed item—and statistically advantageous: for $m$-set combinatorial bandits it improves the minimax rate from $\widetilde{\mathcal{O}}(m\sqrt{KT})$ under full-bandit feedback to $\widetilde{\mathcal{O}}(\sqrt{mKT})$ under semi-bandit feedback (Audibert et al., 2014). In each round, the learner faces either stochastic losses (e.g., from a fixed linear model on random contexts) or adversarial losses (modeling malicious perturbations). Contexts consist of user-feature vectors drawn i.i.d. from a distribution, and instantaneous losses are linear in these features. The best-of-both-worlds (BOBW) paradigm unifies strategies across stochastic and adversarial loss regimes, achieving optimal regret guarantees in both (Zimmert et al., 2019; Tsuchiya et al., 2023). However, gaps remain in current BOBW approaches for contextual combinatorial bandits—highlighted by Tsuchiya et al. (2023); Kuroki et al. (2024) and exacerbated by evolving regulations and emerging technologies—motivating us to revisit the problem.

**Challenges.** While many companies possess extensive user data, their operating environments are rarely stationary and face multiple challenges:

- **Adversarial and Corrupted Stochastic Regimes.** Advertising markets, for instance, evolve in real time as competitors react to each other (Balseiro et al., 2015; Jin et al., 2018), while recommendation systems must adapt to shifting user preferences (Koren, 2009) and defend against malicious actors (Mukherjee et al., 2013; Christakopoulou & Banerjee, 2019; Zhang et al., 2020). A natural remedy is a best-of-both-worlds strategy—provided it is computationally efficient. However, recent privacy regulations (e.g., the EU's GDPR (GDPR, 2016)) restrict context collection (for example, via third-party cookies), so the assumption of perfect knowledge of the context distribution no longer holds. This gap motivates BOBW algorithms that explicitly handle corrupted stochastic contexts—treating unexpected losses as corruptions—to secure optimal regret guarantees, which existing literature has yet to address.

- **Subroutine Computational Efficiency for Combinatorial Bandits.** Advances in large language models (LLMs) allow platforms to generate vast pools of $K$ candidate ads at negligible cost, which can then be deployed for personalized advertising (Meguellati et al., 2024). Yet most existing algorithms require solving a $K$-dimensional convex program in each round, making them increasingly expensive as $K$ grows. Moreover, on-the-fly customization of ad attributes (e.g., color, layout) via generative models further increases the runtime of these subroutines. Consequently, accelerating the per-round projection step is essential for practical, large-scale deployment.

**Contributions.** Our key contributions are summarized as follows.

- **Best-of-Both-Worlds for Contextual Combinatorial Semi-Bandits.** We propose an algorithm for general contextual combinatorial semi-bandits that achieves both $\widetilde{\mathcal{O}}(\sqrt{T})$ regret in the adversarial regime and $\widetilde{\mathcal{O}}(\ln T)$ regret in the corrupted stochastic regime.[1] Our method instantiates Follow-the-Regularized-Leader (FTRL) with a Shannon-entropy regularizer, enabling efficient implementations in many practical settings (see Sec. 2).

- **Accelerated Projection for FTRL/OSMD.** For a popular class of combinatorial action sets, namely the $m$-set, we accelerate the projection subroutine in any FTRL scheme with a Legendre regularizer—equivalently, in Online Stochastic Mirror Descent (OSMD) for linear payoffs under the corresponding mirror map (Shalev-Shwartz, 2012). By exploiting the Karush-Kuhn-Tucker (KKT) conditions, we reduce the typical $K$-dimensional convex projection to a one-dimensional root-finding problem (see Sec. 3). Results therein are generalizable to a broader class of combinatorial action sets that admit a separable structure, *e.g.*, separable matroids.

As a result, our algorithm is comparable to the per-iteration complexity of *Follow-the-Perturbed-Leader* (FTPL) (Neu, 2015; Neu & Bartók, 2016)—which injects random noise to cumulative losses and selects the action with minimal perturbed total loss—while preserving the tight adversarial and stochastic regret guarantees characteristic of FTRL. Hence, we achieve both statistical and computational efficiency.

## 1.1 RELATED WORK

Our work builds on three streams of literature: (i) *contextual combinatorial bandits*, (ii) algorithms that exploit *semi-bandit feedback*, and (iii) *adversarial linear bandits and best-of-both-worlds algorithms*.

**(i) Contextual Combinatorial Bandits.** The contextual combinatorial bandit problem was introduced by Qin et al. (2014), who proposed the $C^2$UCB algorithm for multi-item recommendations. This framework builds on earlier combinatorial semi-bandit models motivated by influence maximization in social networks Chen et al. (2013). Under the i.i.d. reward assumption, linear generalization across arms reduces the regret dependence on the action-set size $K$ from $\sqrt{K}$ to the context dimension $d$. Consequently, when $d \ll K$, the regret scales with $\sqrt{dT}$ instead of $\sqrt{KT}$, offering a substantial statistical advantage. Stochastic variants have explored Thompson sampling Wang & Chen (2018) and refined confidence sets Takemura et al. (2021); however, none of these algorithms provide optimal regret guarantees against an adaptive adversary.

---

[1]Here, $\widetilde{\mathcal{O}}(\cdot)$ suppresses poly-logarithmic factors.

Table 1: Comparison of regret bounds across different settings.

| | Our Paper | Qin et al. (2014) | Zierahn et al. (2023) | Ito et al. (2022) | Kong et al. (2023) |
|---|---|---|---|---|---|
| **Feedback** | Semi-bandit | Full-bandit | Semi/Full-bandit | Graph bandit | Linear bandit |
| **Adv. Regret** | $\widetilde{\mathscr{O}}\left(\text{poly}(d,m,K)\sqrt{T}\right)$ | N/A | $\widetilde{\mathscr{O}}\left(\text{poly}(d,m,K)\sqrt{T}\right)$ | $\widetilde{\mathscr{O}}\left(\sqrt{\alpha T}\right)$ | $\widetilde{\mathscr{O}}\left(\sqrt{T}\right)$ |
| **Stoch. Regret** | $\mathscr{O}\left(\text{poly}(d,m,K)(\ln T)^3\right)$ | $\widetilde{\mathscr{O}}\left(d\sqrt{mT}\right)$ | N/A | $\mathscr{O}\left(\frac{\alpha(\ln T)^3}{\Delta_{\min}}\right)$ | $\mathscr{O}\left(\frac{(\ln T)^2}{\Delta_{\min}}\right)$ |

**(ii) Semi-Bandit Feedback and Efficient Optimization.** Combinatorial bandits were first formalized by Cesa-Bianchi & Lugosi (2012), who generalized EXP3 (Auer et al., 2002) from single arms to binary action vectors. Their algorithm, COMBAND, operates under *full-bandit feedback*—the learner only observes an aggregated loss for the binary action vector they have chosen—and achieves $\widetilde{\mathscr{O}}\left(\sqrt{mKT\ln(K/m)}\right)$ regret. Follow-up work connected this update to mirror descent on combinatorial polytopes: COMPONENTHEDGE Koolen et al. (2010), COMBEXP Combes et al. (2015), and the OSMD analysis of Audibert et al. (2014). Audibert et al. (2014) provide a concise taxonomy and prove matching lower bounds, showing that the $\sqrt{mKT}$ rate is information-theoretically optimal whenever semi-bandit feedback is available. These results are also summarized in (Bubeck & Cesa-Bianchi, 2012, Section 5.6.1). To reduce per-round runtime, Neu & Bartók (2016) introduced a *stochastic* mirror-descent update that samples a single action and updates only the observed coordinates, thereby avoiding the full Bregman projection required by standard OSMD or FTRL. The trade-off is an extra logarithmic factor in the regret bound. Variance-reduced estimators and high-probability analyses later refined these guarantees (Zimmert et al., 2019), yet computing the exact Bregman projection still requires time linear in the number of arms $K$.

**(iii) Adversarial Linear Bandits and Best-of-Both-Worlds.** When losses can adapt to the learner's past, the i.i.d. assumption no longer holds. For single-arm actions ($m = 1$) the classical EXP4 algorithm attains $\widetilde{\mathscr{O}}(\sqrt{T})$ regret, but its running time and memory scale with the number of experts—that is, the total number of deterministic policies mapping contexts to arms—which grows exponentially in the arm size $K$ (Auer et al., 2002). Neu & Olkhovskaya (2020) mitigated this explosion with REALLINEXP3, achieving $\widetilde{\mathscr{O}}(\sqrt{dKT})$ regret under the strong assumption that the context distribution is known. Liu et al. (2023) proposed an algorithm that achieves $\mathscr{O}((dm)^3\sqrt{T})$ regret bound when applied to combinatorial semi-bandit case with only access to 1 context sample per time period $t$, though the resulting log-determinant FTRL remains intractable for the combinatorial bandits case because of exponentially many constraints Foster et al. (2020); Zimmert & Lattimore (2022) and the nonlinearity introduced by their lifting covariance technique. The first near-optimal treatment is the Matrix-Geometric-Resampling (MGR) algorithm of *Zierahn et al.* Zierahn et al. (2023), which achieves $\widetilde{\mathscr{O}}\left(\sqrt{mKT\max\{d,m/\lambda_{\min}(\Sigma)\}}\right)$ regret but fails to provide best-of-both-worlds regret guarantees and relies on a sampling subroutine which requires $\mathscr{O}(\ln T)$ samples in each round. Similar to the issue of generalizing (Liu et al., 2023), a straightforward extension of the best-of-both-worlds (BOBW) algorithm from the linear contextual bandit setting, as studied by Kuroki et al. (2024), to the contextual combinatorial bandit framework yields an action space whose cardinality grows exponentially with the size of the combinatorial decision variables. This exponential growth renders the resulting optimization problem NP-hard and computationally infeasible for large-scale applications. More broadly, the BOBW question—achieving $\widetilde{\mathscr{O}}(\ln T)$ under stochastic contexts and $\widetilde{\mathscr{O}}(\sqrt{T})$ under adversarial ones without prior knowledge—originated in the $K$-armed bandit setting (Bubeck & Slivkins, 2012; Seldin & Slivkins, 2014) and has since been settled for linear bandits via data-dependent stability (Lee et al., 2021), and a streamlined FTRL scheme (Kong et al., 2023) and for contextual bandits via Exp4-style expert reductions (Pacchiano et al., 2022; Dann et al., 2023), at the cost of exponential policy-enumeration. In the purely combinatorial semi-bandit setting, hybrid-regularizer methods (Zimmert et al., 2019; Ito, 2021) achieve the optimal BOBW rates but do not consider contextual information and the stemming difficulty of inverse-covariance matrix estimation. Consequently, no prior method simultaneously handles unknown covariance, large/infinite policy classes, and combinatorial action structure while maintaining BOBW regret bounds. We close this gap with a Shannon-entropy FTRL algorithm that requires neither policy enumeration nor known $\Sigma$, yet still achieves optimal BOBW rates on arbitrary combinatorial action sets.

## 2 BEST-OF-BOTH-WORLDS ALGORITHM FOR CONTEXTUAL COMBINATORIAL SEMI-BANDITS

We begin our analysis of contextual combinatorial bandits by describing the interaction protocol that the learner follows and the problem settings that the best-of-both-worlds framework unifies. Given $K$ base arms, $m$ maximum number of base arms allowed to pull per round, an action set $\mathscr{A} \subseteq \{A \in \{0,1\}^K : \sum_{k=1}^{K}(A)_k \le m\}$, and a context space $\mathscr{X} \subset \mathbb{R}^d$, the interaction protocol for the contextual combinatorial bandit problem proceeds as follows.

**Interaction Protocol.** In each round $t = 1, \ldots, T$ the interaction protocol proceeds as follows. The environment first chooses the loss coefficients $\theta_{t,1}, \ldots, \theta_{t,K} \in \mathbb{R}^d$, and draws an i.i.d. context $X_t \sim \mathscr{D}$. The learner observes $X_t$ and plays the vector $A_t \in \mathscr{A}$. The learner then observes the arm-wise losses $\ell_t(X_t, k) = \langle X_t, \theta_{t,k} \rangle$ for every $k$ such that $(A_t)_k = 1$ and suffers the total loss $\sum_{k=1}^{K} \ell_t(X_t, k)(A_t)_k$. Consistent with the prior work on linear contextual bandits such as (Kuroki et al., 2024), we adopt the following assumption throughout this paper.

**Assumption 1.** *The distribution $\mathscr{D}$, from which contexts $X$ are independently drawn, satisfies $\mathbb{E}[XX^\top] = \Sigma \succ 0$; $\|X\|_2 \le 1$ $\mathscr{D}$-almost surely; $\|\theta_{t,k}\|_2 \le 1$ for all $k \in [K]$ and $t \in [T]$; $\ell_t(x,k) = \langle x, \theta_{t,k} \rangle \in [-1,1]$ for all $x \in \mathscr{X}, k \in [K]$, and $t \in [T]$.*

Under this assumption, we distinguish between the adversarial and stochastic regimes as follows. In the *adversarial regime*, the unknown loss coefficients $\{\theta_{t,k}\}_{k=1}^{K}$ may vary arbitrarily with respect to time but are independent of the learner's action sequence. In the *stochastic regime*, the loss function takes the form $\ell_t(X_t, k) = \langle X_t, \theta_k \rangle + \varepsilon(X_t, k)$ for all $k \in [K]$, where $\theta_k$ is also unknown but fixed throughout the $T$ interaction periods, and $\varepsilon(X_t, k)$ is an independent zero-mean noise. There is an additional intermediate regime that interpolates between the adversarial and stochastic regime, referred to as *corrupted stochastic regime* (Zimmert & Seldin, 2021; Kuroki et al., 2024). In this case, the loss function associated with each base arm $k$ is defined by $\ell_t(X_t, k) = \langle X_t, \theta_{t,k} \rangle + \varepsilon(X_t, k)$, where $\varepsilon(X_t, k)$ is independent zero-mean noise. On the other hand, the coefficients $\theta_{t,k}$ are such that there exist fixed and unknown vectors $\theta_1, \ldots, \theta_K$ and that satisfy $\sum_{t=1}^{T} \max_{A \in \mathscr{A}} \sum_{k=1}^{K} \|\theta_{t,k} - \theta_k\|_2 (A)_k \le C$ for a fixed corruption level constant $C > 0$. Note that $C = 0$ corresponds to the stochastic regime and $C = \Theta(T)$ corresponds to the adversarial regime with additional zero-mean noise. For any fixed context vector $x \in \mathscr{X}$, we define $u^\star : \mathscr{X} \to \mathscr{A}$ as the optimal context-dependent action map that achieves minimum loss in hindsight, that is,

$$u^\star(x) = \operatorname*{argmin}_{A \in \mathscr{A}} \mathbb{E}\left[\sum_{t=1}^{T}\sum_{k=1}^{K} \ell_t(x,k)(A)_k\right].$$

The learner's performance is then measured by the pseudo-regret

$$\mathscr{R}_T = \mathbb{E}\left[\sum_{t=1}^{T}\sum_{k=1}^{K} \ell_t(X_t,k)\left((A_t)_k - \left(u^\star(X_t)\right)_k\right)\right],$$

where the expectation is taken with respect to the action-selection distribution chosen by the learner and the sequence of random contexts and loss coefficients chosen by the environment. As noted in the introduction, it is well-established in the literature that the optimal regret bounds are $\widetilde{\mathscr{O}}(\ln T)$ in the stochastic regime and $\widetilde{\mathscr{O}}(\sqrt{T})$ in the adversarial regime. Algorithms that achieve both rates simultaneously and without prior knowledge of the environment's nature are referred to as "best-of-both-worlds" (Bubeck & Slivkins, 2012; Seldin & Slivkins, 2014).

In the stochastic regime, we define the suboptimality gap associated with an arbitrary action $A \in \mathscr{A}$ and a fixed context $x \in \mathscr{X}$ through $\Delta_A(x) = \sum_{k=1}^{K} \langle x, \theta_k \rangle ((A)_k - (u^\star(x))_k)$, and its minimum as $\Delta_{\min} = \min_{x \in \mathscr{X}} \min_{A \in \mathscr{A} \setminus \{u^\star(x)\}} \Delta_A(x)$. Additionally, we denote $\mathscr{F}_t = \sigma(X_1, A_1, \ldots, X_t, A_t)$ as the $\sigma$-algebra generated by the history of contexts and actions up to and including time $t$. For any positive semi-definite matrix $M \in \mathbb{R}^{d \times d}$, we denote by $\lambda_{\min}(M)$ its smallest eigenvalue. With these definitions and notation in place, we now turn to our proposed method. Algorithm 1 consists of an action-selection rule and a loss-estimation procedure. The action-selection rule applies entropy-regularized FTRL to the convex hull of the action set, then samples an action from the original combinatorial action space, whose cardinality grows exponentially with $m$. Namely, we denote the Shannon entropy

---

**Algorithm 1** FTRL for contextual combinatorial semi-bandits

---

**Require:** Context dimension $d$, subset size $m \leq K$, exploration set $E \subseteq \mathscr{A}$, learning rates $\{\eta_t\}_{t=1}^T$, $\{\alpha_t\}_{t=1}^T$, $\{M_t\}_{t=1}^T$, initialization $\tilde{\theta}_{0,k} = 0$ for all $k \in [K]$
1: **for** $t = 1, \ldots, T$ **do**
2:      Observe context $X_t \in \mathbb{R}^d$
3:      Compute $\bar{A}_t(X_t) \in \operatorname{argmin}_{a \in \operatorname{conv}(\mathscr{A})} \sum_{s=1}^{t-1} \sum_{k=1}^K \langle X_t, \tilde{\theta}_{s,k} \rangle a_k + \psi_t(a)$
4:      Find a distribution $p_t(\cdot|X_t)$ over $\mathscr{A}$ such that $\mathbb{E}_{a \sim p_t(\cdot|X_t)}[a] = \bar{A}_t(X_t)$
5:      Set $\pi_t(a|X_t) = (1 - \alpha_t \eta_t) p_t(a|X_t) + \alpha_t \eta_t \mathbf{1}[a \in E]/|E|$ for all $a \in \mathscr{A}$ and sample $A_t \sim \pi_t(\cdot|X_t)$
6:      Observe loss $\ell_t(X_t, k)$ for all $k \in [K]$ such that $(A_t)_k = 1$
7:      **Subroutine** Precision-matrix estimation$(\pi_t, M_t)$
8:          **for** $n = 1, \ldots, M_t$ **do**
9:              Draw $X(n) \sim \mathscr{D}$ and $A(n) \sim \pi_t(\cdot|X(n))$
10:            Compute $C_{n,k} = \prod_{j=1}^n (I - (A(j))_k X(j) X(j)^\top / 2)$ for all $k \in [K]$
11:          **return** $\widehat{\Sigma}_{t,k}^+ = (I + \sum_{n=1}^{M_t} C_{n,k})/2$ for all $k \in [K]$
12:      **End Subroutine**
13:      Compute $\tilde{\theta}_{t,k} = \widehat{\Sigma}_{t,k}^+ X_t \ell_t(X_t, k)(A_t)_k$ for all $k \in [K]$

---

$H : \operatorname{conv}(\mathscr{A}) \to \mathbb{R}$ by $H(A) = -\sum_{k=1}^K (A)_k \ln(A)_k$ and specify the time-varying regularizer used in Step 3 of Algorithm 1 as $\psi_t(A) = -H(A)/\eta_t$.

The loss estimation is particularly relevant for the contextual case, as learning the optimal action hinges on a computationally tractable approximation of the unknown parameter $\theta_{t,k}$ governing the loss. Given the covariance matrix $\Sigma_{t,k} = \mathbb{E}[(A_t)_k X_t X_t^\top | \mathscr{F}_{t-1}]$, it is known that we can construct the unbiased estimator $\hat{\theta}_{t,k}$ defined through $\hat{\theta}_{t,k} = \Sigma_{t,k}^{-1} X_t \ell_t(X_t, k)(A_t)_k$ for all $k \in [K]$. However, computing this estimator is computationally inefficient as its construction requires computing the inverse of the $d \times d$ covariance matrix $\Sigma_{t,k}$, which is of complexity $\mathscr{O}(d^3)$. Furthermore, this estimation approach assumes that the covariance matrix is known in advance, which is not the case in most real-world scenarios. To avoid such practical problems, we consider relying on the approach of the Matrix-Geometric-Resampling method proposed by (Neu & Olkhovskaya, 2020), as described by the subroutine within Algorithm 1. This approach improves the computational efficiency by a factor on the order of $\mathscr{O}(d)$ compared to computing the naïve unbiased estimator and does not require full knowledge of the context distribution $\mathscr{D}$, at the cost of introducing an additional bias in the estimation of the precision matrix $\Sigma_{t,k}^{-1}$. Note that we only require $M_t = \lceil 4K \ln(t)/(\alpha_t \eta_t \lambda_{\min}(\Sigma)) \rceil = \mathscr{O}(K \ln(T))$ context samples at every time step. Additionally, an exploration set $E \subseteq \mathscr{A}$ is introduced to bound $\lambda_{\min}(\Sigma_{t,k})$ away from zero. The set $E$ thus must satisfy that, for every $k \in [K]$, there is at least one action $A \in E$ such that $(A)_k = 1$. For simplicity, we select $E = \{A \in \{0,1\}^K : \sum_{k=1}^K (A)_k = 1\}$, in which case $|E| = K$. For every $t \in [T]$, we specify the remaining algorithmic parameters as $\eta_t = 1/\beta_t$, where $\beta_t = \max\{2, c_2 \ln T, \beta_t'\}$ and $\beta_{t+1}' = \beta_t' + c_1(1 + (m \ln(K/m))^{-1} \sum_{s=1}^t H(\bar{A}_s(X_s)))^{-1/2}$. We also let $\alpha_t = 4K \ln(t)/\lambda_{\min}(\Sigma)$. In addition, we set the problem-dependent constants $c_1 = \sqrt{(d + \ln T/\lambda_{\min}(\Sigma))K \ln T/(m \ln(K/m))}$ as well as $c_2 = 8K/\lambda_{\min}(\Sigma)$. For initializations, we choose $\beta_1' = c_1 \geq 1$. These definitions ensure that $0 \leq \alpha_t \eta_t \leq 1/2$ and $0 < \eta_t \leq 1/2$ throughout $t = 1, \ldots, T$ rounds.

We state our main results in Theorem 2.1 and sketch the key elements of its proof in the next section. The regret upper bounds presented in Theorem 2.1 are optimal in the dependence on $T$ up to logarithmic factors and, to the best of our knowledge, constitute the first known best-of-both-worlds results for contextual combinatorial semi-bandits. Note that in the corrupted stochastic regime, the corruption budget $C$ enters only as an additive constant in the regret bound (see Appendix A.1 for details) and is therefore subsumed by the $\mathscr{O}(\cdot)$ notation.

**Theorem 2.1** (Best-of-both-worlds regret guarantee for contextual combinatorial bandits). *The regret of Algorithm 1 satisfies the following.*

     *(i) In the adversarial regime, we have $\mathscr{R}_T = \mathscr{O}\left(m\sqrt{K \ln(K/m)T \ln T (d + \ln T/\lambda_{\min}(\Sigma))}\right)$;*

*(ii) In the stochastic regime and the corrupted stochastic regime, we have $\mathscr{R}_T = \mathcal{O}\left(\frac{K \ln T m^{3/2} \ln((K-m)T)(d + \ln T/\lambda_{\min}(\Sigma))}{\Delta_{\min}}\right)$.*

## 2.1 REGRET ANALYSIS

Establishing a best-of-both-worlds guarantee with a computationally efficient algorithm poses several challenges. First, the approach of *Zierahn et al.* (Zierahn et al., 2023), which uses fixed learning and sampling rates yields only an $\widetilde{\mathcal{O}}(\sqrt{T})$ suboptimal regret bound in the stochastic regime—a result of its constant-learning-rate schedule and the coarse penalty bound in the FTRL analysis. Second, adopting the context-less best-of-both-worlds analysis for combinatorial semi-bandits by *Zimmert et al.* (Zimmert et al., 2019) relies on a hybrid regularizer to control arm-wise entropy; this, however, demands arm-wise bias control that state-of-the-art precision-matrix estimators cannot guarantee under standard assumptions. To overcome these limitations, we exploit the time-varying learning rate to refine the regret analysis to be compatible with the stochastic regime, and we lift the mean-action space (support size $K$) to the space of action distributions (support size exponential in $m$). Together, these approaches enable a self-bounding argument commonly used to achieve a $\widetilde{\mathcal{O}}(\ln T)$ regret bound in the stochastic and corrupted stochastic regimes.

Analyzing regret in the contextual combinatorial semi-bandit setting presents an inherent challenge due to the evolving dependency between the sequence of observed contexts $X_1, \ldots, X_T$, and the learned parameters $\{\tilde{\theta}_{1,k}\}_{k=1}^K, \ldots, \{\tilde{\theta}_{T,k}\}_{k=1}^K$. To address this, we follow a strategy inspired by (Neu & Olkhovskaya, 2020; Zierahn et al., 2023), in which we introduce an auxiliary game that simplifies the regret analysis. We define an auxiliary regret notion by introducing a *ghost context sample $X_0 \sim \mathscr{D}$*, drawn independently from the data used to construct the estimates $\{\tilde{\theta}_{t,k}\}_{k=1}^K$. The regret in this auxiliary game is then

$$\widetilde{\mathscr{R}}_T(X_0) = \mathbb{E}\left[\sum_{t=1}^T \sum_{k=1}^K \langle X_0, \tilde{\theta}_{t,k}\rangle \left((A_t)_k - (u^\star(X_0))_k\right)\right].$$

This fixed-context formulation decouples the randomness in the context sequence from the randomness in parameter estimation, making the analysis more tractable. The following result relates the regret in the original contextual semi-bandit setting (as defined in the previous section) to the regret in the auxiliary game.

**Lemma 2.2** (Original game vs. auxiliary game (Neu & Olkhovskaya, 2020, Equation (6))). *For any $X_0 \sim \mathscr{D}$, the regret of Algorithm 1 satisfies*

$$\mathscr{R}_T \leq \mathbb{E}\left[\widetilde{\mathscr{R}}_T(X_0)\right] + 2\sum_{t=1}^T \mathbb{E}\left[\max_{A \in \mathscr{A}} \mathbb{E}\left[\sum_{k=1}^K \langle X_t, \hat{\theta}_{t,k} - \tilde{\theta}_{t,k}\rangle(A)_k \mid \mathscr{F}_{t-1}\right]\right].$$

This decomposition allows us to break down the regret into two components: the *auxiliary regret*, which captures the performance of the algorithm in a fixed-context game, and the *excess bias-induced regret*, which reflects the cumulative effect of using $\tilde{\theta}_{t,k}$ rather than the unbiased estimator $\hat{\theta}_{t,k}$. By analyzing these two terms separately, we can control the total regret under both stochastic and adversarial assumptions.

As an initial step in our regret analysis, we state the following lemma, which bounds the per-round excess regret introduced by the bias in the precision-matrix estimation subroutine. This result enables us to bound the total bias-induced regret in Lemma 2.2 by $\mathcal{O}(\ln T)$ and to control the additional exploration regret in the auxiliary game.

**Lemma 2.3** (Bias control). *For all $t \in [T]$, the estimates $\{\tilde{\theta}_{t,k}\}_{k=1}^K$ constructed in Algorithm 1 satisfy $\max_{A \in \mathscr{A}} \mathbb{E}\left[\sum_{k=1}^K \langle x, \hat{\theta}_{t,k} - \tilde{\theta}_{t,k}\rangle(A)_k \mid \mathscr{F}_{t-1}\right] \leq m/t^2$ for all $x \in \mathscr{X}$.*

We proceed to bound the regret for the auxiliary game, whose proof strategy follows by an FTRL analysis with a carefully-chosen learning-rate schedule, while taking context into account.

**Lemma 2.4** (Regret decomposition for the auxiliary game). *The regret of Algorithm 1 evaluated in the auxiliary game satisfies*

$$\mathbb{E}\left[\widetilde{\mathscr{R}}_T(X_0)\right] \leq L\sqrt{\left(Kd\ln T + \frac{\sqrt{m}K(\ln T)^2}{\lambda_{\min}(\Sigma)}\right)\sum_{t=1}^T \mathbb{E}[H(\bar{A}_t(X_0))]} + \frac{8Km\ln(K/m)\ln T}{\lambda_{\min}(\Sigma)},$$

*where $L > 0$ is a universal constant.*

Lemma 2.4 serves as the combinatorial bandits analogue of Lemma 3 in (Kuroki et al., 2024), which was established for linear contextual bandits. However, while (Kuroki et al., 2024) defines entropy over the space of action distributions, our setting defines entropy over the mean-action polytope. As a result, directly generalizing their entropy bound to settings with multiple arm pulls is insufficient for establishing an optimal regret bound in the stochastic regime for contextual combinatorial bandits. To overcome these limitations, we derive a refined entropy bound by exploiting a careful partitioning of the base arm set $[K]$ and lifting the mean-action space (support size $K$) to the space of action distributions (support size exponential in $m$). This refined bound enables the application of the self-bounding technique from (Zimmert & Seldin, 2021) in the stochastic setting.

**Lemma 2.5** (Refined entropy bound in the stochastic regime). *Take a ghost sample $X_0 \sim \mathscr{D}$ and any action sequence $A_1, \ldots, A_T$ generated under the policy sequence $\pi_1, \ldots, \pi_T$ using Algorithm 1. Suppose that $\sum_{t=1}^{T} \sum_{k:(u^\star(X_0))_k=0} (A_t)_k \geq e$, where $e$ denotes Euler's number. The mean-action sequence $\bar{A}_1, \ldots, \bar{A}_T$ generated by Algorithm 1 then satisfies*

$$\mathbb{E}\left[\sum_{t=1}^{T} H(\bar{A}_t(X_0))\right] \leq m \ln((K-m)T) \mathbb{E}\left[\sum_{t=1}^{T} \sum_{A \in \mathscr{A} \setminus \{u^\star(X_t)\}} \pi_t(A|X_t)\right].$$

The proof for Theorem 2.1 then follows by combining the insights of all lemmas presented in this section. The complete proof and auxiliary results are deferred to Appendix A.1.

## 3   EFFICIENT NUMERICAL SCHEME FOR COMBINATORIAL SEMI-BANDITS

In the adversarial combinatorial semi-bandit setting, as presented in Section 2, the learner must perform FTRL/OSMD updates over the convex hull of the combinatorial action space—an operation that naively entails a $K$-dimensional Bregman projection. By exploiting the KKT conditions of this convex subproblem, we reduce each update to a single one-dimensional root-finding call per round, yielding a more computationally efficient scheme. We first formalize the interaction protocol and then present the OSMD algorithm (Algorithm 2) under the context-free regime. For simplicity, we study in this section the $m$-set setting, where exactly $m$ arms are pulled in each round. Note that in terms of computing Bregman projection, the context-free $m$-set setting is without loss of generality, since Step 3 of Algorithm 1 is computed under a fixed context and we may iterate through $m$ number of $j$-subsets, where $j = 1, \ldots, m$. An efficient $\widetilde{\mathscr{O}}(K)$ action-sampling procedure in the $m$-set setting is proposed in (Zimmert et al., 2019, Appendix B.2).

**Interaction Protocol.** Fix the number of base arms $K$ and $m$-set size $m \leq K$. Let $\mathscr{A} = \{A \in \{0,1\}^K : \sum_{k=1}^{K}(A)_k = m\}$. Then, in each round $t = 1, \ldots, T$, the interaction protocol proceeds as follows. The environment chooses an adversarial loss vector $\ell_t = (\ell_{t,1}, \ldots, \ell_{t,K}) \in [-1,1]^K$. The learner then plays the vector $A_t \in \mathscr{A}$, suffers the total loss $\langle A_t, \ell_t \rangle = \sum_{k=1}^{K} \ell_{t,k}(A_t)_k$, and observes the coordinate losses $\ell_{t,k}$ for every $k$ such that $(A_t)_k = 1$.

---

**Algorithm 2** Online stochastic mirror descent for semi-bandits (Lattimore & Szepesvári, 2020, Algorithm 18)

---

**Require:** $m$-set size $m \leq K$, learning rate $\eta$, function $F$
 1: Initialize $\bar{A}_1 = \arg\min_{a \in \text{conv}(\mathscr{A})} F(a)$
 2: **for** $t = 1, \ldots, T$ **do**
 3:     Choose distribution $p_t$ over $\mathscr{A}$ such that $\mathbb{E}_{a \sim p_t}[a] = \bar{A}_t$
 4:     Sample action $A_t \sim p_t$, observe partial losses $\ell_{t,k}$ for all $k$ with $(A_t)_k = 1$
 5:     Compute the importance-weighted loss estimator $\hat{\ell}_{t,k} = (A_t)_k \ell_{t,k}/(\bar{A}_t)_k$ for all $k \in [K]$
 6:     Update the decision vector by solving $\bar{A}_{t+1} = \arg\min_{a \in \text{conv}(\mathscr{A})} \{\eta \langle a, \hat{\ell}_t \rangle + D_F(a, \bar{A}_t)\}$

---

Algorithm 2 proceeds in each round by: (i) sampling an action $A_t \sim p_t$, (ii) computing the importance-weighted loss estimator $\hat{\ell}_{t,k}$, and (iii) updating the mean action via the Bregman projection. Here, $D_F(a, \bar{A}_t)$ is the Bregman divergence from Definition 1.

**Definition 1** (Bregman divergence). *Given a convex differentiable function $F : \mathrm{conv}(\mathscr{A}) \to \mathbb{R}$, the Bregman divergence $D_F : \mathrm{conv}(\mathscr{A}) \times \mathrm{conv}(\mathscr{A}) \to \mathbb{R}_+$ associated with $F$ is defined as*

$$D_F(a, \bar{A}) = F(a) - F(\bar{A}) - \langle \nabla F(\bar{A}), a - \bar{A} \rangle \quad \forall a, \bar{A} \in \mathrm{conv}(\mathscr{A}).$$

Assuming that the convex potential function $F : \mathrm{conv}(\mathscr{A}) \to \mathbb{R}$ is separable, that is, $F$ can be expressed as $F(a) = \sum_{k=1}^{K} f(a_k)$ where $f : \mathbb{R} \to \mathbb{R}$ is convex, and $a_k$ is the $k$-th coordinate of $a \in \mathscr{A}$. Hence, the update step amounts to solving the convex subproblem

$$\min_{a \in \mathrm{conv}(\mathscr{A})} \eta \langle a, \hat{\ell}_t \rangle + D_F(a, \bar{A}_t), \tag{1}$$

whose unique minimizer we denote by $a^\star$.

We derive the KKT conditions as the following. Let us assign the following Lagrangian multipliers $\lambda \in \mathbb{R}$ to the constraint $\sum_{k=1}^{K} a_k = m$, then $\mu \in \mathbb{R}^K$ to the set of constraints $a_k \geq 0$, $\forall k \in [K]$ and $\nu \in \mathbb{R}^K$ to the set of inequality constraints $a_k \leq 1$, $\forall k \in [K]$. Then the Lagrangian has the form $\mathscr{L}(a, \lambda, \mu, \nu) = \sum_{k=1}^{K} [\eta \hat{\ell}_{t,k} a_k + f(a_k) - f((\bar{A}_t)_k) - f'((\bar{A}_t)_k)(a_k - (\bar{A}_t)_k)] + \lambda(\sum_{k=1}^{K} a_k - m) - \sum_{k=1}^{K} \mu_k a_k + \sum_{k=1}^{K} \nu_k(a_k - 1)$. The resulting KKT conditions for equation 1 are as follows.

$$
\begin{aligned}
0 \leq a_k \leq 1, \ \sum_{k=1}^{K} a_k = m, \ \forall k \in [K] \quad &\text{(Primal feasibility)} \\
\mu_k \geq 0, \ \nu_k \geq 0, \ \forall k \in [K] \quad &\text{(Dual feasibility)} \\
\mu_k a_k = 0, \ \nu_k(a_k - 1) = 0, \ \forall k \in [K] \quad &\text{(Complementary slackness)} \\
\eta \hat{\ell}_{t,k} + f'(a_k) - f'((\bar{A}_t)_k) + \lambda - \mu_k + \nu_k = 0, \ \forall k \in [K] \quad &\text{(Stationarity)}
\end{aligned}
$$

We continue to reformulate the stationarity condition by distinguishing between cases for an arbitrary arm index $k \in [K]$. First, note that when $a_k > 0$, complementary slackness implies that $\mu_k = 0$. Substituting into the stationarity condition yields $f'(a_k) + \lambda + c_k = 0$, where we use the shorthand notation $c_k = \eta \hat{\ell}_{t,k} - f'((\bar{A}_t)_k)$. Inverting $f'$ then gives $(f')^{-1}(-\lambda - c_k) = a_k$. Next, consider the case when $a_k = 0$, the stationarity condition together with dual feasibility $\mu_k \geq 0$ gives $f'(0) + \lambda + c_k = \mu_k \geq 0$, which implies $(f')^{-1}(-\lambda - c_k) \leq 0$. Since the range of $(f')^{-1}$ is $[0, 1]$, we conclude that $(f')^{-1}(-\lambda - c_k) = 0 = a_k$.

Thus, we have shown that solving the $K$-dimensional convex optimization problem equation 1 reduces to a one-dimensional root-finding problem $\sum_{k=1}^{K} (f')^{-1}(-\lambda - c_k) = m$.

---

**Algorithm 3** Bisection algorithm for solving equation 1

---

**Require:** Tolerance $\varepsilon$, loss parameters $c$
1: Initialize $\underline{\lambda} = \min_{k \in [K]} \{-c_k - f'(m/K)\}$ and $\bar{\lambda} = \max_{k \in [K]} \{-c_k - f'(m/K)\}$
2: **for** $l = 1$ to $\log_2(\varepsilon^{-1} 2L\sqrt{K}(\bar{\lambda} - \underline{\lambda}))$ **do**
3: $\quad \lambda = (\underline{\lambda} + \bar{\lambda})/2$
4: $\quad$ **if** $m - \sum_{k=1}^{K} (f')^{-1}(-\lambda - c_k) > 0$ **then**
5: $\quad\quad \bar{\lambda} \leftarrow \lambda$
6: $\quad$ **else**
7: $\quad\quad \underline{\lambda} \leftarrow \lambda$
8: **return** $a_k = m/K + (f')^{-1}(-\underline{\lambda} - c_k) - (1/K)\sum_{j=1}^{K} (f')^{-1}(-\underline{\lambda} - c_j)$ for all $k \in [K]$

---

Similar to the result in (Li et al., 2024, Theorem 6.1), which analyzes a perturbation-based algorithm specialized for multi-armed bandits, our bisection algorithm enjoys the following convergence guarantee.

**Theorem 3.1** (Convergence of Algorithm 3). *Suppose that $f$ is strictly convex and differentiable, and that $(f')^{-1}$ is $L$-Lipschitz continuous. Then, for any tolerance $\varepsilon > 0$, Algorithm 3 outputs $a \in \mathrm{conv}(\mathscr{A})$ with $\sum_{k=1}^{K} a_k = m$ and $\|a - a^\star\|_2 \leq \varepsilon$.*

Even when $(f')^{-1}$ is not available in closed form, we may resort to some approximation oracle. Below we discuss how the convergence proof could be modified to accommodate the approximation error.

**Corollary 3.2** (Convergence with an approximate inverse oracle). *Assume we have an oracle that, on input $z \in \mathbb{R}$, returns $\tilde{y} = (f')^{-1}(z)$ satisfying $|\tilde{y} - (f')^{-1}(z)| \le \tau$ for some known tolerance $\tau > 0$. Then, under the same assumptions as in Theorem 3.1, and provided that $\tau \le \varepsilon/(2\sqrt{K})$, the approximate algorithm yields a vector $\tilde{a} \in \mathrm{conv}(\mathscr{A})$ satisfying $\|\tilde{a} - a^\star\|_2 \le \varepsilon$ in $\mathcal{O}(\ln(L\sqrt{K}(\bar{\lambda} - \underline{\lambda})/\varepsilon))$ bisection iterations.*

Note that Algorithm 2 calls Algorithm 3 with input $c_k = \eta \hat{\ell}_{t,k} - f'((\bar{A}_t)_k)$ for all $k \in [K]$ in each iteration $t = 1, \ldots, T$ in order to compute the mean-action vector $\bar{A}_t$. Thus, the width of the search interval $\bar{\lambda} - \underline{\lambda}$ is on the order of $\mathcal{O}(t)$ with high probability. This observation implies that the $t$-th call to Algorithm 3 requires $\mathcal{O}(\ln(\sqrt{K}\eta T))$ iterations with high probability. In addition, each iteration runs in time $\mathcal{O}(K)$. Hence, if $\eta = \mathcal{O}(1/\sqrt{T})$, then the $t$-th call of Algorithm 3 runs in time at most $\mathcal{O}(K \ln(\sqrt{KT})) = \widetilde{\mathcal{O}}(K)$ with high probability. Using the sampling scheme of complexity $\widetilde{\mathcal{O}}(K)$ proposed by Zimmert et al. (2019), the efficiency of Algorithm 3 as used by Algorithm 2 is thus comparable to the sampling procedure employed by FTPL (Neu & Bartók, 2016).

We remark on the extension of our results in this section to settings beyond $m$-set such as a partition matroid (Oxley, 2011). Although we now have cardinality constraints per partition instead of a single cardinality constraint, each partition is handled by its own scalar root-finding problem, and every arm enters exactly one partition. Hence, the work across all partitions sums to $K$, which is the same as the uniform-matroid case. Building on the previous observation, the Bregman projection using our method still runs in time $O(K \ln(1/\varepsilon))$; the bisection method over partition $i$ runs in time $O(c_i \ln(1/\varepsilon))$ with $K = \sum_i c_i$.

### 3.1 NUMERICAL EXPERIMENTS

We now evaluate the per-iteration runtime of Algorithm 3. All experiments are conducted on a machine with a 2.3 GHz 8-core Intel Core i9 processor and all optimization problems are modeled in Python. In all experiments, we fix the $m$-set size to $m = 5$, vary the number of base arms $K \in \{10, \ldots, 100\}$, and run each algorithm for $N = 25$ iterations on a loss vector $y \in [0,1]^K$ whose entries are drawn uniformly at random. Mean per-iteration runtimes and their 95% confidence intervals are reported, and two instances of problem equation 1, each employing a different regularizer, are evaluated. The first instance uses *Tsallis entropy* with parameter $\alpha = 1/2$, as in (Zimmert et al., 2019; Zimmert & Seldin, 2021), which is known to achieve best-of-both-worlds results. This corresponds to the regularizer $f(x) = -\sqrt{x}$ (labeled as "Tsallis"). The second instance employs the widely used negative *Shannon entropy*, induced by the regularizer $f(x) = x \ln x$ (labeled as "Negative Shannon entropy").

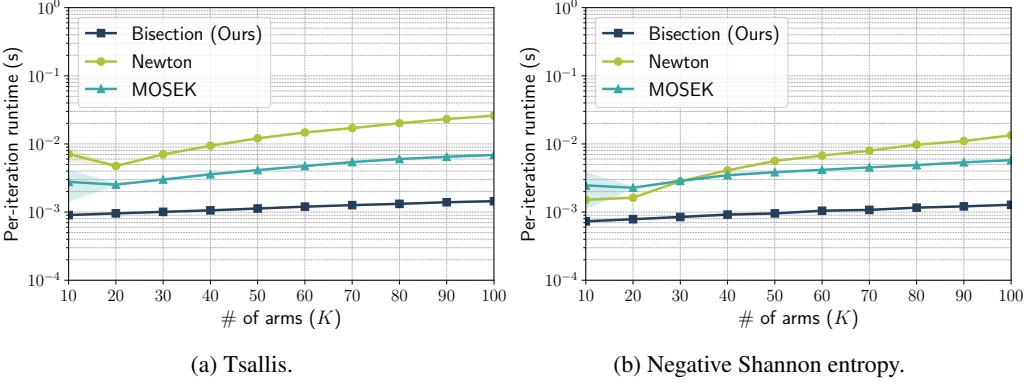

(a) Tsallis.

(b) Negative Shannon entropy.

Figure 1: Per-iteration runtime for different regularizers.

We compute the mean-action projection of FTRL via Algorithm 3, and compare it against two baselines: the heuristic Newton method used in (Zimmert et al., 2019), [2] and a direct implementation

---

[2]The source code for this method is available at `https://github.com/diku-dk/CombSemiBandits`.

of the optimization step using MOSEK.[3] In all cases, we solve to an error tolerance of $\varepsilon = 10^{-7}$, matching the suboptimality and feasibility tolerances used in MOSEK. Figure 1 visualizes the per-iteration runtimes of Algorithm 3, the Newton method, and MOSEK, as a function of the number $K$ of base arms. We observe that Algorithm 3 runs nearly 10 times faster than the Newton baseline for $K = 100$, and consistently outperforms MOSEK by a factor of approximately 5 across all values of $K$. This highlights the computational efficiency of our bisection-based Algorithm 3.

## 4    CONCLUDING REMARKS AND LIMITATIONS

We introduce the first algorithm for *contextual combinatorial semi-bandits* that simultaneously guarantees $\widetilde{\mathcal{O}}(\sqrt{T})$ regret in the adversarial regime and $\widetilde{\mathcal{O}}(\ln T)$ in the corrupted stochastic regime. At its core is a *Follow-the-Regularized-Leader* (FTRL) scheme with a Shannon-entropy regularizer with time-varying learning rate. Vanilla FTRL, however, requires solving a $K$-dimensional convex projection in each round, which limits scalability, whereas *Follow-the-Perturbed-Leader* (FTPL) is favored for its speed. We recover FTPL-style speed-ups by exploiting the KKT conditions to reduce the projection to a one-dimensional root-finding problem. This hybrid design preserves FTRL's strong theoretical guarantees while matching FTPL's per-round efficiency. Despite these advances, our approach has two main limitations. First, the Shannon-entropy regularizer introduces an additional $\widetilde{\mathcal{O}}(\ln T)$ term in the adversarial regret. Second, the dependence on the combinatorial action size $m$ scales as $\sqrt{m}$, which is suboptimal relative to known lower bounds (Zierahn et al., 2023). Addressing the aforementioned limitations would be a promising direction for further enhancing both the theory and practice of best-of-both-worlds bandit algorithms.

## ACKNOWLEDGEMENTS

This work was supported as part of the NCCR Automation, a National Center of Competence in Research, funded by the Swiss National Science Foundation (grant number 51NF40_225155).

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

# A APPENDIX

## A.1 PROOFS AND AUXILIARY RESULTS FOR SECTION 2

For ease of notation in the proofs, we set $\gamma_t = \alpha_t \eta_t$ throughout this section. We start by addressing the terms relevant to the regret induced in the auxiliary game for a fixed context $\widetilde{\mathscr{R}}_T(x)$. Denote as well the time-varying Bregman divergence from $p \in \text{conv}(\mathscr{A})$ to $q \in \text{conv}(\mathscr{A})$ through

$$D_t(q, p) = \psi_t(q) - \psi_t(p) - \langle \nabla \psi_t(p), q - p \rangle.$$

We start by the following lemma by following a standard FTRL analysis with varying potentials.

**Lemma A.1** (Stability-penalty decomposition). *For any context $x \in \mathscr{X}$ and $\gamma_t \leq 1$ for all $t \in [T]$, we have*

$$\mathbb{E}_{A_t} \left[ \sum_{t=1}^{T} \sum_{k=1}^{K} \langle x, \tilde{\theta}_{t,k} \rangle ((A_t)_k - (u^\star(x))_k) \right]$$

$$\leq \underbrace{\sum_{t=1}^{T} \left( \psi_t(\bar{A}_{t+1}(x)) - \psi_{t+1}(\bar{A}_{t+1}(x)) \right) + \psi_{T+1}(u^\star(x)) - \psi_1(\bar{A}_1(x))}_{\text{Penalty}}$$

$$+ \underbrace{\sum_{t=1}^{T} (1 - \gamma_t) \sum_{k=1}^{K} \langle x, \tilde{\theta}_{t,k} \rangle \left( (\bar{A}_t(x))_k - (\bar{A}_{t+1}(x))_k \right) - D_t(\bar{A}_{t+1}(x), \bar{A}_t(x)) +}_{\text{Stability}} \underbrace{U(x)}_{\text{Exploration-induced regret}},$$

(2)

*where $U(x) = \sum_{t=1}^{T} \gamma_t \sum_{k=1}^{K} \langle x, \tilde{\theta}_{t,k} \rangle (1/|E| - (u^\star(x))_k)$.*

*Proof of Lemma A.1.* It follows from the construction of $\bar{A}_t(x)$ that

$$\mathbb{E}_{A_t} \left[ \sum_{t=1}^{T} \sum_{k=1}^{K} \langle x, \tilde{\theta}_{t,k} \rangle \left( (A_t)_k - (u^\star(x))_k \right) \right]$$

$$= \sum_{t=1}^{T} \sum_{k=1}^{K} (1 - \gamma_t) \left( (\bar{A}_t(x))_k - (u^\star(x))_k \right) \langle x, \tilde{\theta}_{t,k} \rangle + \sum_{t=1}^{T} \sum_{k=1}^{K} \gamma_t \left( \frac{1}{|E|} - (u^\star(x))_k \right) \langle x, \tilde{\theta}_{t,k} \rangle.$$

Applying a standard FTRL regret decomposition result (see e.g. (Lattimore & Szepesvári, 2020, Exercise 28.12)) to the first term on the right-hand side of the above expression and applying the definition of $U(x)$ yields the desired result. $\square$

The following lemma is a building block for bounding the stability term appearing in Lemma A.1.

**Lemma A.2** (Bound for the scaled per-arm loss). *Under the assumptions of Lemma A.1, we have*

$$\max_{k \in [K]} |\eta_t \langle x, \tilde{\theta}_{t,k} \rangle| \leq 1.$$

*Proof of Lemma A.2.* Observe that

$$\max_{k \in [K]} |\eta_t \langle x, \tilde{\theta}_{t,k} \rangle| = \max_{k \in [K]} \left| \eta_t \langle x, \widehat{\Sigma}_{t,k}^+ X_t \ell_t(X_t, k)(A_t)_k \rangle \right|$$

$$\leq \max_{k \in [K]} \left| \eta_t x^\top \widehat{\Sigma}_{t,k}^+ X_t \right|$$

$$\leq \eta_t \max_{k \in [K]} \| \widehat{\Sigma}_{t,k}^+ \|_{\text{op}} \leq \frac{\eta_t (M_t + 1)}{2} \leq \frac{\eta_t \left( \frac{4K \ln(t)}{\alpha_t \eta_t \lambda_{\min}(\Sigma)} + 1 \right)}{2} \leq 1,$$

where the second inequality exploits Hölder's inequality, which applies because $\|X_t\|_2 \leq 1$ by assumption, the third inequality holds thanks to (Zierahn et al., 2023, Lemma 1), the fourth inequality holds due to the parameter choice $\gamma_t = \alpha_t \eta_t$. The last inequality again follows by the parameter choices $\alpha_t = 4K \ln(t)/\lambda_{\min}(\Sigma)$ and $\eta_t \leq 1/2$. $\square$

We now decompose the regret for the auxiliary game stated in Lemma A.1 as follows.

**Lemma A.3** (Regret breakdown for the auxiliary game). *Under the assumptions of Lemma A.1, we have*

$$\widetilde{\mathscr{R}}_T(x) - U(x) \leq \sum_{t=1}^{T} (1 - \gamma_t) \eta_t e \sum_{k=1}^{K} \langle x, \tilde{\theta}_{t,k} \rangle^2 (\bar{A}_t(x))_k + \sum_{t=1}^{T} (\beta_{t+1} - \beta_t) H(\bar{A}_{t+1}(x)) + mc_2 \ln(K/m) \ln T.$$

*Proof of Lemma A.3.* We analyze the stability and penalty terms appearing in Lemma A.1. First, we bound the per-round stability term in equation 2. By (Lattimore & Szepesvári, 2020, Theorem 26.13) and using that $\frac{\partial^2}{(\partial x)^2}\psi_t(x) = \frac{1}{\eta_t x}$ for all $x \in \text{conv}(\mathscr{A})$, we have

$$\sum_{k=1}^{K}\langle x,\tilde{\theta}_{t,k}\rangle\big((\bar{A}_t(x))_k - (\bar{A}_{t+1}(x))_k\big) - D_t(\bar{A}_{t+1}(x),\bar{A}_t(x)) \leq \frac{\eta_t}{2}\sum_{k=1}^{K}\langle x,\tilde{\theta}_{t,k}\rangle^2(z_t)_k, \qquad (3)$$

where $z_t$ lies on the line segment connecting $\bar{A}_t(x)$ and $q^\star \in \text{argmax}_{q\in\mathbb{R}^K}\sum_{k=1}^{K}\langle x,\tilde{\theta}_{t,k}\rangle((\bar{A}_t(x))_k - q_k) - D_t(q,\bar{A}_t(x))$. In addition, by the first-order optimality conditions we have $q_k^\star = (\bar{A}_t(x))_k\exp(-\eta_t\langle x,\tilde{\theta}_{t,k}\rangle)$. Combining the previous observation with Lemma A.2 which states that $-\eta_t\langle x,\tilde{\theta}_{t,k}\rangle \in [-1,1]$ for all $k \in [K]$, we deduce that $q_k^\star \in [(\bar{A}_t(x))_k/e, e(\bar{A}_t(x))_k]$. Because $z_t$ lies on the line segment connecting $\bar{A}_t$ and $q^\star$, we then have $(z_t)_k \leq e(\bar{A}_t(x))_k$ for all $k \in [K]$, which in turn implies that

$$\sum_{k=1}^{K}\langle x,\tilde{\theta}_{t,k}\rangle^2(z_t)_k \leq e\sum_{k=1}^{K}\langle x,\tilde{\theta}_{t,k}\rangle^2(\bar{A}_t(x))_k. \qquad (4)$$

Substituting equation 4 into equation 3 establishes the upper bound for the stability term as claimed. As for the penalty term, we have

$$\sum_{t=1}^{T}\big(\psi_t(\bar{A}_{t+1}(x)) - \psi_{t+1}(\bar{A}_{t+1}(x))\big) + \psi_{T+1}(u^\star(x)) - \psi_1(\bar{A}_1(x))$$

$$\leq \sum_{t=1}^{T}(\beta_{t+1} - \beta_t)H(\bar{A}_{t+1}(x)) + \frac{m}{\eta_1}\ln(K/m) \leq \sum_{t=1}^{T}(\beta_{t+1} - \beta_t)H(\bar{A}_{t+1}(x)) + mc_2\ln(K/m)\ln T,$$

where the first inequality holds because of Jensen's inequality and noting that $H$ takes nonnegative values on $\text{conv}(\mathscr{A})$, and the second inequality holds because of the choice $\beta_1 = \max\{2, c_2\ln T, c_1\}$. Thus, the claim follows. $\qquad\square$

We continue to bound the extra regret for the auxiliary game due to exploration, which uses Lemma 2.3 as a building block. For the sake of completeness we first state the proof of Lemma 2.3.

*Proof of Lemma 2.3.* (Zierahn et al., 2023, Lemma 3) and our choice of $E$ that satisfies $|E| = K$ imply that

$$\max_{A\in\mathscr{A}}\mathbb{E}\left[\sum_{k=1}^{K}\langle x,\hat{\theta}_{t,k} - \tilde{\theta}_{t,k}\rangle(A)_k \mid \mathscr{F}_{t-1}\right] \leq \sqrt{m}\exp\left(-\frac{\gamma_t\lambda_{\min}(\Sigma)M_t}{2K}\right) \quad \forall x \in \mathscr{X}.$$

The claim then follows by the choice of $M_t$ used in Algorithm 1. $\qquad\square$

We are now equipped with the technical tools necessary to bound the exploration-induced regret, as stated below.

**Lemma A.4** (Extra regret due to exploration). *We have $\mathbb{E}[U(X_0)] \leq (\sqrt{m}+1)\mathbb{E}\left[\sum_{t=1}^{T}\gamma_t\right]$.*

*Proof of Lemma A.4.* Observe that

$$\mathbb{E}[U(X_0)] \leq \mathbb{E}\left[\sum_{t=1}^{T}\gamma_t\max_{A\in\mathscr{A}}\mathbb{E}\left[\sum_{k=1}^{K}\langle X_0,\tilde{\theta}_{t,k} - \hat{\theta}_{t,k} + \hat{\theta}_{t,k}\rangle(A)_k \mid \mathscr{F}_{t-1}\right]\right]$$

$$\leq \mathbb{E}\left[\sum_{t=1}^{T}\gamma_t\max_{A\in\mathscr{A}}\mathbb{E}\left[\sum_{k=1}^{K}\langle X_0,\tilde{\theta}_{t,k} - \hat{\theta}_{t,k}\rangle(A)_k + \ell_t(X_0,k) \mid \mathscr{F}_{t-1}\right]\right]$$

$$\leq \mathbb{E}\left[\sum_{t=1}^{T}\gamma_t\max_{A\in\mathscr{A}}\mathbb{E}\left[\sum_{k=1}^{K}\langle X_0,\tilde{\theta}_{t,k} - \hat{\theta}_{t,k}\rangle(A)_k + 1 \mid \mathscr{F}_{t-1}\right]\right] \leq (\sqrt{m}+1)\mathbb{E}\left[\sum_{t=1}^{T}\gamma_t\right],$$

where the second inequality follows by the unbiasedness of $\hat{\theta}_{t,k}$, the third inequality holds by the assumption that $\ell_t(x,k) \leq 1$ for all $x \in \mathscr{X}$ and $k \in [K]$, and the last inequality holds because of Lemma 2.3. $\qquad\square$

The following is another preparation lemma for establishing Lemma 2.4, which constitutes an adaptation of (Kuroki et al., 2024, Lemma 18) to the combinatorial semi-bandit setting. It provides a refined bound on the penalty term induced by the Shannon entropy regularizer with respect to the ghost sample.

**Lemma A.5** (Entropic bound for the ghost sample). *We have*

$$\mathbb{E}\left[\sum_{t=1}^{T}(\beta'_{t+1}-\beta'_t)H(\bar{A}_{t+1}(X_0))\right] = \mathcal{O}\left(c_1\sqrt{m\ln(K/m)}\sqrt{\sum_{t=1}^{T}\mathbb{E}[H(\bar{A}_t(X_0))]}\right).$$

*Proof of Lemma A.5.* By definition of $\beta'_t$, we obtain

$$\mathbb{E}\left[\sum_{t=1}^{T}(\beta'_{t+1}-\beta'_t)H(\bar{A}_{t+1}(X_0))\right] = \mathbb{E}\left[\sum_{t=1}^{T}\frac{c_1}{\sqrt{1+(m\ln(K/m))^{-1}\sum_{s=1}^{t}H(\bar{A}_s(X_s))}}H(\bar{A}_{t+1}(X_0))\right]$$

$$\leq 2c_1\sqrt{m\ln(K/m)}\mathbb{E}\left[\sum_{t=1}^{T}\frac{H(\bar{A}_{t+1}(X_0))}{\sqrt{\sum_{s=1}^{t+1}H(\bar{A}_s(X_s))}+\sqrt{\sum_{s=1}^{t}H(\bar{A}_s(X_s))}}\right],$$

where in the last step we used the fact that $H(\bar{A}_s(X_s)) \leq m\ln(K/m)$. The above upper bound further reduces to

$$2c_1\sqrt{m\ln(K/m)}\mathbb{E}\left[\sum_{t=1}^{T}\frac{H(\bar{A}_{t+1}(X_0))}{\sqrt{\sum_{s=1}^{t+1}H(\bar{A}_s(X_s))}+\sqrt{\sum_{s=1}^{t}H(\bar{A}_s(X_s))}}\right]$$

$$= 2c_1\sqrt{m\ln(K/m)}\mathbb{E}\left[\sum_{t=1}^{T}\frac{H(\bar{A}_{t+1}(X_0))(\sqrt{\sum_{s=1}^{t+1}H(\bar{A}_s(X_s))}-\sqrt{\sum_{s=1}^{t}H(\bar{A}_s(X_s))})}{H(\bar{A}_{t+1}(X_0))}\right]$$

$$= 2c_1\sqrt{m\ln(K/m)}\mathbb{E}\left[\sum_{t=1}^{T}\sqrt{\sum_{s=1}^{t+1}H(\bar{A}_s(X_s))}-\sqrt{\sum_{s=1}^{t}H(\bar{A}_s(X_s))}\right]$$

$$= 2c_1\sqrt{m\ln(K/m)}\mathbb{E}\left[\sqrt{\sum_{s=1}^{T+1}H(\bar{A}_s(X_s))}-\sqrt{H(\bar{A}_1(X_1))}\right]$$

$$\leq 2c_1\sqrt{m\ln(K/m)}\mathbb{E}\left[\sqrt{\sum_{s=1}^{T}H(\bar{A}_s(X_s))}\right] \leq c_1\sqrt{m\ln(K/m)}\sqrt{\sum_{s=1}^{T}\mathbb{E}[H(\bar{A}_s(X_s))]},$$

where the first equality holds because $\mathbb{E}_{X_{t+1}\sim\mathscr{D}}\left[H(\bar{A}_{t+1}(X_{t+1}))|\mathscr{F}_t\right] = \mathbb{E}_{X_0\sim\mathscr{D}}\left[H(\bar{A}_{t+1}(X_0))|\mathscr{F}_t\right]$. The first inequality exploits the fact that $H(\bar{A}_s(X_s)) \leq H(\bar{A}_1(X_1)) = m\ln(K/m)$, and the second inequality follows from Jensen's inequality. Thus, the claim follows. $\square$

The following lemma provides an upper bound on the refined per-round stability term established in Lemma A.3 in the form of variance of the parameter estimates $\{\tilde{\theta}_{t,k}\}_{k=1}^{K}$ for all $t = 1,\dots,T$.

**Lemma A.6** (Variance control (Zierahn et al., 2023, Lemma 5)). *For any context $x \in \mathscr{X}$, conditioning on the history $\mathscr{F}_{t-1}$ yields the variance bound*

$$(1-\gamma_t)\mathbb{E}\left[\sum_{k=1}^{K}\langle x,\tilde{\theta}_{t,k}\rangle^2(\bar{A}_t(x))_k \mid \mathscr{F}_{t-1}\right] \leq 3Kd.$$

We are now equipped with all the technicalities needed to establish the regret bound for the original game.

*Proof of Lemma 2.4.* We begin by establishing an upper bound on the sum of learning rates, $\sum_{t=1}^{T}\eta_t$, which will be useful throughout the proof. Observe first that by construction of $\beta'_t$, we have

$$\beta'_t = c_1 + \sum_{s=1}^{t-1}\frac{c_1}{\sqrt{1+(m\ln(K/m))^{-1}\sum_{u=1}^{s-1}H(\bar{A}_u(X_u))}} \geq \frac{c_1 t}{\sqrt{1+(m\ln(K/m))^{-1}\sum_{s=1}^{t}H(\bar{A}_s(X_s))}},$$

where the inequality holds because $H(\bar{A}_u(X_u)) \geq 0$ for all $u \in [t]$. Thus,

$$\sum_{t=1}^{T} \eta_t \leq \sum_{t=1}^{T} \frac{1}{\beta_t'} \leq \sum_{t=1}^{T} \frac{\sqrt{1 + (m\ln(K/m))^{-1} \sum_{s=1}^{t} H(\bar{A}_s(X_s))}}{c_1 t}$$

$$\leq \frac{1 + \ln T}{c_1} \sqrt{1 + (m\ln(K/m))^{-1} \sum_{s=1}^{T} H(\bar{A}_s(X_s))} \qquad (5)$$

$$= \mathcal{O}\left( \frac{\ln T}{c_1 \sqrt{m\ln(K/m)}} \sqrt{\sum_{t=1}^{T} H(\bar{A}_t(X_t))} \right),$$

where we used $H(\bar{A}_1(X_1)) = m\ln(K/m)$. It then follows that

$$\mathbb{E}\left[ \sum_{t=1}^{T} (1 - \gamma_t) \eta_t e \sum_{k=1}^{K} \langle x, \tilde{\theta}_{t,k} \rangle^2 (\bar{A}_t(x))_k \right] = \mathbb{E}\left[ \sum_{t=1}^{T} \eta_t e \mathbb{E}\left[ (1 - \gamma_t) \sum_{k=1}^{K} \langle x, \tilde{\theta}_{t,k} \rangle^2 (\bar{A}_t(x))_k \mid \mathscr{F}_{t-1} \right] \right]$$

$$\leq \mathcal{O}\left( \mathbb{E}\left[ \frac{Kd \cdot \ln T}{c_1 \sqrt{m\ln(K/m)}} \sqrt{\sum_{t=1}^{T} H(\bar{A}_t(X_t))} \right] \right) \qquad (6)$$

$$\leq \mathcal{O}\left( \frac{Kd \cdot \ln T}{c_1 \sqrt{m\ln(K/m)}} \sqrt{\mathbb{E}\left[ \sum_{t=1}^{T} H(\bar{A}_t(X_t)) \right]} \right),$$

where the equality holds due to the law of iterated expectations, the first inequality holds thanks to Lemma A.6 and equation 5, and the second inequality follows from Jensen's inequality.

We continue to bound the extra regret due to exploration by the cumulative entropy term. It follows from Lemma A.4 that

$$\mathbb{E}[U(X_0)] \leq (\sqrt{m} + 1) \mathbb{E}\left[ \sum_{t=1}^{T} \gamma_t \right] \leq (\sqrt{m} + 1) \mathbb{E}\left[ \sum_{t=1}^{T} \frac{4\eta_t K \ln T}{\lambda_{\min}(\Sigma)} \right]$$

$$= \mathcal{O}\left( \frac{K(\ln T)^2}{c_1 \lambda_{\min}(\Sigma) \sqrt{\ln(K/m)}} \sqrt{\sum_{t=1}^{T} \mathbb{E}[H(\bar{A}_t(X_t))]} \right), \qquad (7)$$

where the second inequality holds because $\gamma_t = \alpha_t \eta_t$ and the choice of $\alpha_t$ as well as the fact that $\ln t \leq \ln T$. The equality then holds because of equation 5.

Finally, we establish an upper bound for the penalty term in the regret decomposition. Denote by $t_0$ the first round in which $\beta_t'$ becomes larger than the constant $F = \max\{2, c_2 \ln T\}$, i.e., $t_0 = \min\{t \in [T] : \beta_t' \geq F\}$. We then have

$$\mathbb{E}\left[ \sum_{t=1}^{T} (\beta_{t+1} - \beta_t) H(\bar{A}_{t+1}(X_0)) \right]$$

$$= \mathbb{E}\left[ \sum_{t=1}^{t_0-1} (\beta_{t+1} - \beta_t) H(\bar{A}_{t+1}(X_0)) + \sum_{t=t_0}^{T} (\beta_{t+1} - \beta_t) H(\bar{A}_{t+1}(X_0)) \right]$$

$$\leq \mathbb{E}\left[ (\beta_{t_0}' - \beta_{t_0-1}') H(\bar{A}_{t+1}(X_0)) + \sum_{t=t_0}^{T} (\beta_{t+1}' - \beta_t') H(\bar{A}_{t+1}(X_0)) \right] \qquad (8)$$

$$\leq \mathbb{E}\left[ \sum_{t=1}^{T} (\beta_{t+1}' - \beta_t') H(\bar{A}_{t+1}(X_0)) \right] = \mathcal{O}\left( c_1 \sqrt{m\ln(K/m)} \sqrt{\sum_{t=1}^{T} \mathbb{E}[H(\bar{A}_t(X_0))]} \right),$$

where the first inequality is due to the fact that $\beta_t = \beta_{t+1}$ while $t \in [t_0 - 2]$, $\beta_{t_0-1} \geq \beta_{t_0-1}'$ by construction, and $\beta_t' = \beta_t$ for $t \geq t_0$. The second inequality holds because $\beta_t'$ is increasing in $t$, while the second equality holds thanks to Lemma A.5. The claim then follows by substituting equation 6, equation 7, and equation 8 into terms in the statement of Lemma A.3 (to bound the regret for the auxiliary game) as well as Lemma 2.2 (to bound the regret for the original game). $\qquad \square$

The crux to show the best-of-both-worlds result now lies in constructing a tight upper bound on the cumulative entropy term that dominates the regret bound in Lemma 2.4.

*Proof of Lemma 2.5.* Observe that for any $A \in \text{conv}(\mathscr{A})$ and any $S \subset [K]$ with $|S| \leq m$, it follows that

$$
\begin{aligned}
H(A) &= \sum_{k \notin S} (A)_k \ln \frac{1}{(A)_k} + \sum_{k \in S} (A)_k \ln \frac{1}{(A)_k} \\
&\leq \sum_{k \notin S} (A)_k \ln \frac{K - |S|}{\sum_{k \notin S} (A)_k} + \sum_{k \in S} (A)_k \left( \frac{1}{(A)_k} - 1 \right) \\
&= \sum_{k \notin S} (A)_k \left( \ln \frac{K - |S|}{\sum_{k \notin S} (A)_k} + 1 \right),
\end{aligned}
$$

where the inequality holds because of Jensen's inequality and because $\ln(1/x) \leq 1/x - 1$, and the second equality holds because $\sum_{k \in S}(A)_k + \sum_{k \notin S}(A)_k = |S|$ thanks to the membership of $A$ in $\text{conv}(\mathscr{A})$. Applying the above inequality to the entropy with respect to action given the ghost sample $X_0$ and $S = \{k \in [K] : (u^\star(X_0))_k = 1\}$ gives

$$
\begin{aligned}
\sum_{t=1}^{T} H(\bar{A}_t(X_0)) &\leq \sum_{t=1}^{T} \sum_{k:(u^\star(X_0))_k=0} (\bar{A}_t)_k \left( \ln \frac{K - |S|}{\sum_{k:(u^\star(X_0))_k=0}(\bar{A}_t)_k} + 1 \right) \\
&\leq \sum_{t=1}^{T} \sum_{k:(u^\star(X_0))_k=0} (\bar{A}_t)_k \ln \frac{e(K - |S|)T}{\sum_{t=1}^{T} \sum_{k:(u^\star(X_0))_k=0}(\bar{A}_t)_k} \qquad (9) \\
&\leq \ln((K - |S|)T) \sum_{t=1}^{T} \sum_{k:(u^\star(X_0))_k=0} (\bar{A}_t)_k,
\end{aligned}
$$

where the second inequality follows by Jensen's inequality, and the third inequality holds because

$$
\sum_{t=1}^{T} \sum_{k:(u^\star(X_0))_k=0} (\bar{A}_t)_k \geq e.
$$

We continue to bound the term $\sum_{t=1}^{T} \sum_{k:(u^\star(X_0))_k=0} (\bar{A}_t)_k$. Taking expectations yields

$$
\begin{aligned}
\sum_{t=1}^{T} \sum_{k:(u^\star(X_0))_k=0} (\bar{A}_t)_k &= \sum_{t=1}^{T} \sum_{k:(u^\star(X_0))_k=0} \sum_{A \in \mathscr{A}} \pi_t(A|X_t)(A)_k \\
&= \sum_{t=1}^{T} \sum_{k:(u^\star(X_0))_k=0} \sum_{A \in \mathscr{A} \setminus \{u^\star(X_0)\}} \pi_t(A|X_t)(A)_k \\
&\leq |S| \sum_{t=1}^{T} \sum_{A \in \mathscr{A} \setminus \{u^\star(X_0)\}} \pi_t(A|X_t).
\end{aligned}
$$

Substituting the above expression into equation 9, noticing that $\mathbb{E}[\pi_t(u^\star(X_0))|\mathscr{F}_{t-1}] = \mathbb{E}[\pi_t(u^\star(X_t))|\mathscr{F}_{t-1}]$ and that $|S| \leq m$ thus establishes the claim. $\qquad\square$

Finally, Theorem 2.1 can be established via combining all of the insights yielded in the above lemmas.

*Proof of Theorem 2.1.* For ease of notation, let us denote $\kappa = \sqrt{Kd \ln T + K(\ln T)^2 / \lambda_{\min}(\Sigma)}$. The regret bound stated in (i) follows immediately from Lemma 2.2 combined with Lemma 2.4 as well as the observation that $\sum_{t=1}^{T} H(\bar{A}_t(X_0)) \leq m \ln(K/m)T$. We now show claim (ii). Note that in the case of $\sum_{t=1}^{T} \sum_{k:(u^\star(X_0))_k=0} (\bar{A}_t)_k < e$, we have $\sum_{t=1}^{T} H(\bar{A}_t(X_0)) \leq e \ln(e(K - m)T) + 1/e$, in which case we already have the desired bound. We thus continue to consider the case when $\sum_{t=1}^{T} \sum_{k:(u^\star(X_0))_k=0} (\bar{A}_t)_k \geq e$. Observe that due to the definition of suboptimality gap $\Delta_A(x)$ and the corruption budget $C$, it

follows that

$$
\begin{aligned}
\mathscr{R}_T &\geq \mathbb{E}\left[\sum_{t=1}^{T}\sum_{A\in\mathscr{A}\setminus\{u^\star(X_t)\}}\pi_t(A|X_t)\Delta_A(x)\right] - 2\mathbb{E}\left[\sum_{t=1}^{T}\max_{A\in\mathscr{A}}\sum_{k=1}^{K}\|X_t\|_2\|\theta_{t,k}-\theta_k\|_2(A)_k\right] \\
&\geq \Delta_{\min}\mathbb{E}\left[\sum_{t=1}^{T}\sum_{A\in\mathscr{A}\setminus\{u^\star(X_t)\}}\pi_t(A|X_t)\right] - 2C.
\end{aligned}
\tag{10}
$$

For any $\lambda \in [0,1]$, we may decompose the regret as

$$
\begin{aligned}
\mathscr{R}_T &= (1+\lambda)\mathscr{R}_T - \lambda\mathscr{R}_T \\
&\leq (1+\lambda)\kappa\sqrt{\mathbb{E}\left[\sum_{t=1}^{T}H(\bar{A}_t(X_0))\right]} - \lambda\Delta_{\min}\mathbb{E}\left[\sum_{t=1}^{T}\sum_{A\in\mathscr{A}\setminus\{u^\star(X_t)\}}\pi_t(A|X_t)\right] + 2\lambda C \\
&\quad + \mathcal{O}\left(\frac{K}{\lambda_{\min}(\Sigma)}m\ln(K/m)\ln T\right) \\
&\leq (1+\lambda)\kappa\sqrt{\ln((K-m)T)}\sqrt{m\mathbb{E}\left[\sum_{t=1}^{T}\sum_{a\in\mathscr{A}\setminus\{u^\star(X_t)\}}\pi_t(A|X_t)\right]} \\
&\quad - \lambda\Delta_{\min}\mathbb{E}\left[\sum_{t=1}^{T}\sum_{A\in\mathscr{A}\setminus\{u^\star(X_t)\}}\pi_t(A|X_t)\right] + \mathcal{O}\left(\frac{K}{\lambda_{\min}(\Sigma)}m\ln(K/m)\ln T\right) \\
&\leq \mathcal{O}\left(\frac{(1+\lambda)^2\kappa^2 m\ln((K-m)T)}{4\lambda\Delta_{\min}}\right) = \mathcal{O}\left(\frac{\kappa^2 m\ln((K-m)T)}{\Delta_{\min}}\right),
\end{aligned}
$$

where the first inequality holds thanks to Lemma 2.2 combined with Lemma 2.4 as well as the lower bound equation 10, the second inequality follows by Lemma 2.5. The third inequality holds by the observation $a\sqrt{x}-bx \leq a^2/(4b)$ for any nonnegative scalars $a,b,x$, and the last equality follows by choosing $\lambda = 1$. $\qquad\square$

## A.2 Proofs and Auxiliary Results for Section 3

**Example 1** (Choices of $f$). *We provide three examples of the arm-wise regularizer $f$ on domain $[0,1]$ or $(0,1]$ that admits a closed-form expression of $(f')^{-1}$ and satisfies the assumptions of Theorem 3.1.*

1. *$f(x) : [0,1] \to \mathbb{R}$ is defined through $f(x) = x\ln x - x$ for $x \in (0,1]$ and $f(0) = 0$, which is continuous on $[0,1]$ and differentiable on $(0,1]$. Note that although $\ln x$ is not defined at $x = 0$, we have $\lim_{x\to 0^+}x\ln x = 0$. Observe that its first derivative for $x \in (0,1]$ is $\ln x$. It then follows that $(f')^{-1}(-\lambda - c_k) = e^{-\lambda - c_k}$ for all $k \in [K]$. In addition, note that $z = f'(x) = \ln x$ ranges over the interval $z \in (-\infty, 0]$. On that interval, the derivative of the inverse $\partial e^z/\partial z = e^z$ satisfies $e^z \leq e^0 = 1$ for all $z \in (-\infty, 0]$. By the mean-value theorem, for any $z_1, z_2 \in [-\infty, 0]$ there exists $c$ between them such that $|e^{z_1} - e^{z_2}| = e^c|z_1 - z_2| \leq 1|z_1 - z_2|$. Hence, the function $(f')^{-1}(-\lambda - c_k) = e^{-\lambda - c_k}$ is 1-Lipschitz.*

2. *$f(x) = x^2$ on $[0,1]$. It then follows that $(f')^{-1}(z) = z/2$ for all $z \in [0,2]$. As in the previous case, with $z = -\lambda - c_k$, for all $k \in [K]$ we have $(f')^{-1}(-\lambda - c_k) = (-\lambda - c_k)/2$. Note also that the function $(f')^{-1}(-\lambda - c_k) = (-\lambda - c_k)/2$ is globally $(1/2)$-Lipschitz on its entire domain.*

3. *$f(x) = -\sqrt{x}$ on $(0,1]$.*

   *Similar to previous calculations, we have $x = (f')^{-1}(z) = 1/(4z^2)$. Note that it must hold $z < 0$ for the expression to be defined properly. Hence, with $z = -\lambda - c_k < 0$, for all $k \in [K]$, we have $(f')^{-1}(-\lambda - c_k) = 1/(4(-\lambda - c_k)^2)$. As $x \in (0,1]$ we have $z = -1/(2\sqrt{x}) \in (-\infty, -1/2]$. On that interval, the derivative of the inverse is:*

   $$
   \frac{\partial}{\partial z}\left(\frac{1}{4z^2}\right) = -\frac{1}{2z^3}.
   $$

*On the interval $z \in \left(-\infty, -\frac{1}{2}\right]$, the largest magnitude of the inverse of the derivative map $|(f')^{-1}(z)|$ occurs at the smallest $|z|$, namely at $|z| = 1/2$. This follows from the fact that $|(f')^{-1}(z)|$ is decreasing as $|z|$ grows. Then, $\sup |(f')^{-1}(z)| = 4$, on $z \in (-\infty, -1/2]$. So, $|(f')^{-1}(z_1) - (f')^{-1}(z_2)| \leq 4|z_1 - z_2|$, for all $z_1, z_2 \in (-\infty, -1/2]$ Therefore, $(f')^{-1}$ is 4-Lipschitz.*

*Proof of Theorem 3.1.* We first show that the endpoints of the search interval give rise to strictly negative and strictly positive function values, respectively. The definition of $\underline{\lambda}$ in Step 1 of Algorithm 3 implies that

$$-f'(m/K) \leq -\underline{\lambda} - c_k \quad \forall k \in [K]. \tag{11}$$

As $f$ is strictly convex, its first derivative $f'$ is strictly increasing, which in turn implies that its inverse $(f')^{-1}$ is also strictly increasing. Applying the inverse function to both sides of equation 11 yields

$$(f')^{-1}(-\underline{\lambda} - c_k) \geq (f')^{-1}\left(f'\left(\frac{m}{K}\right)\right) = \frac{m}{K} \quad \forall k \in [K].$$

Summing over $k = 1, \ldots, K$ gives $\sum_{k=1}^{K}(f')^{-1}(-\underline{\lambda} - c_k) \geq m$. A similar argument applied to $\bar{\lambda}$ gives $\sum_{k=1}^{K}(f')^{-1}(-\bar{\lambda} - c_k) \leq m$. Thus, by the intermediate value theorem, there exists at least one $\lambda^* \in [\underline{\lambda}, \bar{\lambda}]$ such that

$$\sum_{k=1}^{K}(f')^{-1}(-\lambda^\star - c_k) = m.$$

Next, we show that the prescribed iteration number gives an approximate solution of $\lambda^*$ upon termination. By the assumed $L$-Lipschitz continuity of $(f')^{-1}$, we have

$$|(f')^{-1}(x) - (f')^{-1}(y)| \leq L|x - y| \quad \forall x, y \in \mathbb{R}. \tag{12}$$

Define the modulus of uniform continuity $\delta(\varepsilon)$ by

$$\delta(\varepsilon) = \max_{\delta > 0}\{\delta : |(f')^{-1}(x) - (f')^{-1}(y)| \leq \varepsilon/(2\sqrt{K}) \, \forall x, y \in \mathbb{R} \text{ with } |x - y| \leq \delta\}.$$

Choosing $\delta_0 = \varepsilon/(2L\sqrt{K})$ gives $|(f')^{-1}(x) - (f')^{-1}(y)| \leq L\delta_0 = \varepsilon/(2\sqrt{K})$, which implies that $\delta(\varepsilon) \geq \varepsilon/(2L\sqrt{K})$. Let the initial interval length be $\Delta_0 = \bar{\lambda} - \underline{\lambda}$, where $\bar{\lambda}, \underline{\lambda}$ are initializations specified in Step 1 of Algorithm 3. Note that each bisection iteration halves the interval's length. Therefore, the interval length after $l$ iterations is $\Delta_l = \Delta_0/2^l$. Denote by $\lambda_l$ the midpoint of the $l$-th interval. Since the true root $\lambda^*$ lies within this interval, the error satisfies

$$|\lambda_l - \lambda^\star| \leq \frac{\Delta_l}{2} = \frac{\Delta_0}{2^{l+1}}.$$

To guarantee that the propagated error through the $L$-Lipschitz function $(f')^{-1}$ remains below $\varepsilon/(2\sqrt{K})$ in each coordinate, it suffices to have $L|\lambda_l - \lambda^\star| \leq \varepsilon/(2\sqrt{K})$, or equivalently, $\Delta_0/2^{l+1} \leq \varepsilon/(2L\sqrt{K})$. It follows that the bisection algorithm should terminate as soon as

$$l \geq \log_2\left(\frac{2L\sqrt{K}\Delta_0}{\varepsilon}\right) - 1.$$

Let denote with $L^*$ the number of iterations after bisection algorithm terminates. Then,

$$|\lambda_{L^*} - \lambda^\star| \leq \frac{\varepsilon}{2L\sqrt{K}}.$$

and thus $L^* \leq \log_2(2L\sqrt{K}\Delta_0/\varepsilon)$. The output of Algorithm 3 is defined as

$$a_k = (f')^{-1}(-\lambda - c_k) + \frac{\Delta}{K}, \quad \forall k \in [K], \tag{13}$$

where $\Delta = m - \sum_{k=1}^{K}(f')^{-1}(-c_k - \lambda^\star)$. For each coordinate $k$, the error induced by the difference between $\lambda$ and $\lambda^*$ is dominated by the Lipschitz property equation 12 through

$$\left|(f')^{-1}(-\lambda - c_k) - (f')^{-1}(-\lambda^\star - c_k)\right| \leq L|\lambda - \lambda^\star| \leq \frac{\varepsilon}{2\sqrt{K}},$$

which in turn implies that $(f')^{-1}(-\lambda - c_k) \geq (f')^{-1}(-\lambda^\star - c_k) - \varepsilon/(2\sqrt{K})$. Summing over $k = 1,\ldots,K$ and by defining $\Delta_\lambda = m - \sum_{k=1}^{K}(f')^{-1}(-\lambda - c_k)$ yields:

$$\Delta_\lambda \leq m - \left[\sum_{k=1}^{K}(f')^{-1}(-\lambda^\star - c_k) - \frac{\varepsilon\sqrt{K}}{2}\right].$$

By the definition of $\Delta$ we can bound $\Delta_\lambda$ as $\Delta_\lambda \leq \Delta + \varepsilon\sqrt{K}/2$. In the worst-case scenario when the computed error $\Delta_\lambda$ is close to 0, the inequality above reduces to $\Delta \geq \Delta_\lambda - \varepsilon\sqrt{K}/2$. Since $\Delta_\lambda$ is nearly 0 or in the worst-case non-positive, this yields the bound $\Delta \geq -\varepsilon\sqrt{K}/2$. Therefore, from equation 13, the error in each coordinate satisfies $|a_k - a_k^\star| \leq \varepsilon/\sqrt{K}$, and therefore $\|a - a^\star\|_2 \leq \varepsilon$. Thus, the claim follows. $\qquad\square$

*Proof of Corollary 3.2.* Define the exact residual $g(\lambda) = m - \sum_{k=1}^{K}(f')^{-1}(-\lambda - c_k)$ and the approximate residual $\tilde{g}(\lambda) = m - \sum_{k=1}^{K}\tilde{y}_k$, where each $\tilde{y}_k$ satisfies $|\tilde{y}_k - (f')^{-1}(-\lambda - c_k)| \leq \tau$. To measure the oracle's perturbation, set $E(\lambda) = \tilde{g}(\lambda) - g(\lambda) = -\sum_{k=1}^{K}\left[\tilde{y}_k - (f')^{-1}(-\lambda - c_k)\right]$. Since this is a sum of $K$ individual errors, the triangle inequality gives $|E(\lambda)| \leq \sum_{k=1}^{K}|\tilde{y}_k - (f')^{-1}(-\lambda - c_k)| \leq K\tau$.

In the exact inverse case we assume $g(\underline{\lambda}) \geq 0$ and $g(\bar{\lambda}) \leq 0$, which guarantees a root $\lambda^\star \in [\underline{\lambda}, \bar{\lambda}]$. Accounting for the oracle error then yields $\tilde{g}(\underline{\lambda}) \geq -K\tau$ and $\tilde{g}(\bar{\lambda}) \leq K\tau$.

Provided that $\min\{g(\underline{\lambda}), -g(\bar{\lambda})\} > K\tau$, the sign test on $\tilde{g}$ still selects the correct subinterval; hence the bisection converges with each evaluation of $g$ perturbed by at most $\pm K\tau$. After $l$ iterations, the midpoint $\lambda_l$ satisfies $|\lambda_l - \lambda^\star| \leq \Delta_0/2^{l+1}$, with $\Delta_0 = \bar{\lambda} - \underline{\lambda}$. Upon termination at $\lambda_l$, define $\tilde{a}_k = \tilde{y}_k(\lambda_l)$ for all $k \in [K]$. Finally, subtract the mean of $\tilde{a}$ to ensure that $\sum_{k=1}^{K}\tilde{a}_k = m$ exactly.

Subtracting the mean of $\tilde{a}$ introduces a uniform shift of magnitude $C = \mathcal{O}(\tau + L|\lambda_l - \lambda^\star|)$. Hence each coordinate error can be bounded as

$$|\tilde{a}_k - a_k^\star| \leq \underbrace{|(f')^{-1}(-\lambda_l - c_k) - (f')^{-1}(-\lambda^\star - c_k)|}_{\leq L|\lambda_l - \lambda^\star|} + \underbrace{|\tilde{y}_k - (f')^{-1}(-\lambda_l - c_k)|}_{\leq \tau} + C.$$

Since $(f')^{-1}$ is $L$-Lipschitz and each oracle error is bounded by $\tau$, it follows that $|\tilde{a}_k - a_k^\star| \leq L|\lambda_l - \lambda^\star| + \tau + C$. Taking the $\ell_2$-norm of the coordinatewise bound gives

$$\|\tilde{a} - a^\star\|_2 \leq \sqrt{K}\max_i|\tilde{a}_k - a_k^\star| \leq \sqrt{K}\left(L|\lambda_l - \lambda^\star| + \tau + C\right) = \mathcal{O}\left(\sqrt{K}(\tau + L|\lambda_l - \lambda^\star|)\right).$$

Hence, to ensure $\|\tilde{a} - a^\star\|_2 \leq \varepsilon$, it suffices that

$$L|\lambda_l - \lambda^\star| \leq \frac{\varepsilon}{2\sqrt{K}} \quad \text{and} \quad \tau \leq \frac{\varepsilon}{2\sqrt{K}}.$$

The first inequality follows from

$$|\lambda_l - \lambda^\star| \leq \frac{\Delta_0}{2^{l+1}} \leq \frac{\varepsilon}{2\sqrt{K}L},$$

which is equivalent to

$$l \geq \log_2\left(\frac{2\sqrt{K}L\Delta_0}{\varepsilon}\right),$$

while the second is simply $\tau \leq \varepsilon/(2\sqrt{K})$. Together, these conditions imply the claimed iteration bound $\mathcal{O}\left(\ln(L\sqrt{K}(\bar{\lambda} - \underline{\lambda})/\varepsilon)\right)$. This observation completes the proof. $\qquad\square$

# B  ADDITIONAL NUMERICAL EXPERIMENTS

In Sec. 3.1 we reported the per-iteration runtime of Algorithm 3. To further validate both the approach and our implementation, we plot cumulative-regret trajectories for OSMD (Algorithm 2), comparing (a) our Bisection-based projection with (b) a direct solution of the projection step via MOSEK.[4]

---

[4]All code to reproduce the figures is available at `https://github.com/RAO-EPFL/BOBW-CCB`.

Mean cumulative regrets for the stochastic and adversarial settings are shown in Figs. 2, 3, 4, and 5, respectively, under the Tsallis and negative Shannon-entropy regularizers. We consider $m \in \{3,5\}$, $K \in \{20,40\}$ base arms, and a horizon of $T = 10^4$ rounds.

Across all configurations, the two implementations yield indistinguishable regret curves (up to numerical tolerance), supporting the correctness of our implementation. For completeness, Tables 2 and 3 report final cumulative regret for all projection subroutines considered in the paper (Bisection, Newton, MOSEK) under the Tsallis and negative Shannon-entropy regularizers, respectively.

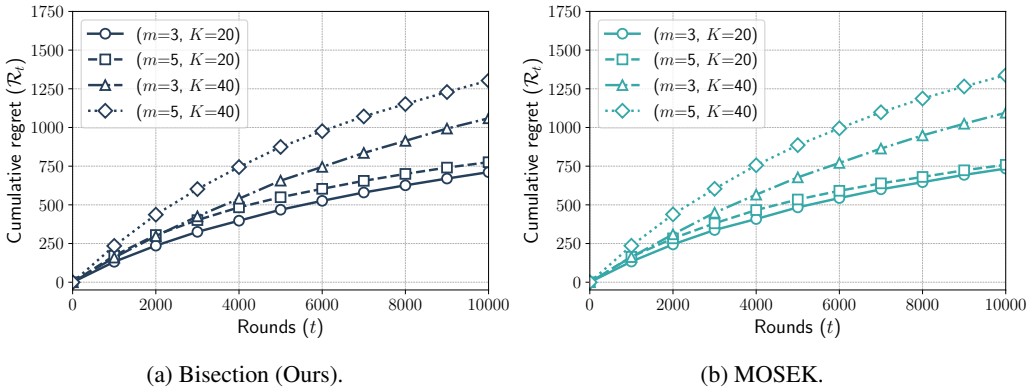

(a) Bisection (Ours).          (b) MOSEK.

Figure 2: Stochastic setting for $\Delta = 0.0625$ (Tsallis regularizer).

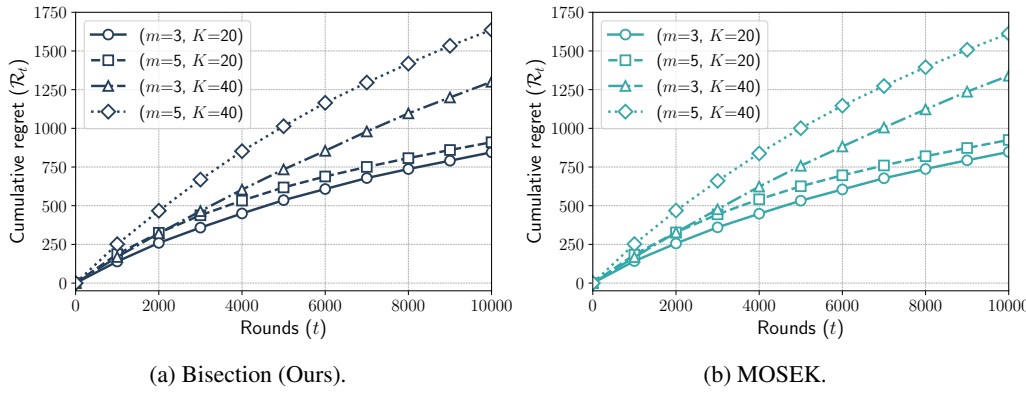

(a) Bisection (Ours).          (b) MOSEK.

Figure 3: Stochastic setting for $\Delta = 0.0625$ (Negative Shannon entropy regularizer).

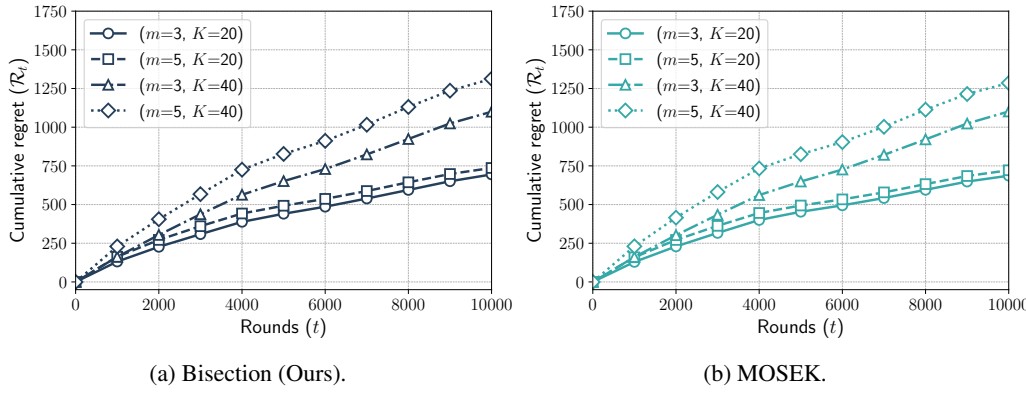

(a) Bisection (Ours).          (b) MOSEK.

Figure 4: Adversarial setting for $\Delta = 0.0625$ (Tsallis regularizer).

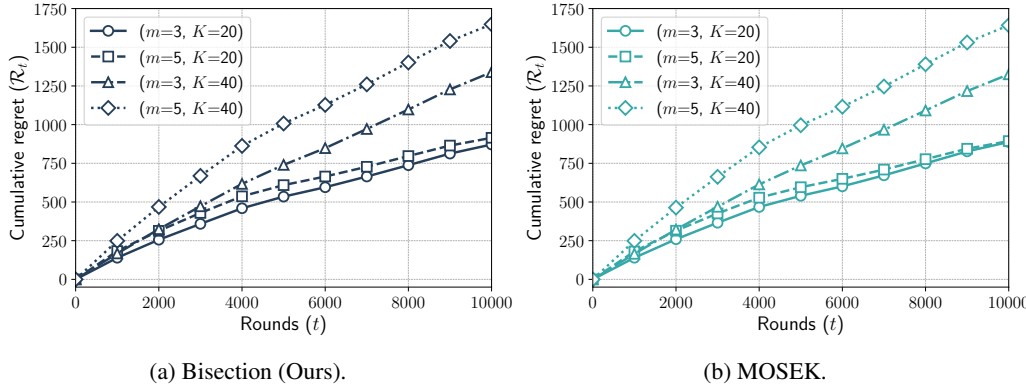

(a) Bisection (Ours).  (b) MOSEK.

Figure 5: Adversarial setting for $\Delta = 0.0625$ (Negative Shannon entropy regularizer).

Table 2: Cumulative regret at $T = 10^4$ for $\Delta = 0.0625$ (Tsallis regularizer).

| $K$ | $m$ | Method | Stochastic ($\mu \pm \sigma$) | Adversarial ($\mu \pm \sigma$) |
|---|---|---|---|---|
| 20 | 3 | Bisection (Ours) | $711.3 \pm 146.3$ | $694.6 \pm 161.2$ |
|  |  | Newton | $726.2 \pm 170.1$ | $710.8 \pm 130.4$ |
|  |  | MOSEK | $734.5 \pm 143.0$ | $686.6 \pm 88.1$ |
|  | 5 | Bisection (Ours) | $775.7 \pm 170.2$ | $736.5 \pm 119.7$ |
|  |  | Newton | $761.2 \pm 164.3$ | $734.7 \pm 121.3$ |
|  |  | MOSEK | $758.8 \pm 170.1$ | $721.5 \pm 137.4$ |
| 40 | 3 | Bisection (Ours) | $1059.4 \pm 190.2$ | $1099.7 \pm 143.4$ |
|  |  | Newton | $1114.8 \pm 181.4$ | $1117.4 \pm 171.5$ |
|  |  | MOSEK | $1095.2 \pm 176.1$ | $1101.4 \pm 119.7$ |
|  | 5 | Bisection (Ours) | $1303.2 \pm 167.3$ | $1312.6 \pm 199.0$ |
|  |  | Newton | $1314.9 \pm 210.2$ | $1309.8 \pm 187.6$ |
|  |  | MOSEK | $1338.0 \pm 197.4$ | $1285.6 \pm 164.9$ |

Table 3: Cumulative regret at $T = 10^4$ for $\Delta = 0.0625$ (Negative Shannon entropy regularizer).

| $K$ | $m$ | Method | Stochastic ($\mu \pm \sigma$) | Adversarial ($\mu \pm \sigma$) |
|---|---|---|---|---|
| 20 | 3 | Bisection (Ours) | $845.1 \pm 160.9$ | $871.5 \pm 107.2$ |
|  |  | Newton | $835.6 \pm 136.9$ | $874.0 \pm 102.8$ |
|  |  | MOSEK | $847.4 \pm 115.7$ | $887.6 \pm 131.4$ |
|  | 5 | Bisection (Ours) | $910.2 \pm 164.4$ | $914.8 \pm 126.2$ |
|  |  | Newton | $914.4 \pm 152.4$ | $900.5 \pm 128.8$ |
|  |  | MOSEK | $925.1 \pm 160.6$ | $893.7 \pm 129.9$ |
| 40 | 3 | Bisection (Ours) | $1300.8 \pm 149.4$ | $1339.2 \pm 115.3$ |
|  |  | Newton | $1281.1 \pm 118.8$ | $1342.8 \pm 101.9$ |
|  |  | MOSEK | $1340.1 \pm 144.5$ | $1325.0 \pm 123.5$ |
|  | 5 | Bisection (Ours) | $1636.6 \pm 193.4$ | $1648.3 \pm 156.4$ |
|  |  | Newton | $1568.7 \pm 168.5$ | $1629.0 \pm 141.8$ |
|  |  | MOSEK | $1612.7 \pm 168.0$ | $1642.0 \pm 170.9$ |

