# OpenReview forum: "Efficient Best-of-Both-Worlds Algorithms for Contextual Combinatorial Semi-Bandits"
_ICLR.cc/2026/Conference — ICLR 2026 Poster_

### Official Review · Reviewer_vDJH · 2025-10-24

**Soundness:** 3
**Presentation:** 2
**Contribution:** 2
**Rating:** 4
**Confidence:** 4

**Summary:**

This paper studies the contextual combinatorial bandit problems and investigates the best-of-both-worlds algorithm that achieves $O(\sqrt{T} )$ regret in the adversarial regime.
and polylog T regret in the stochastic regime. The paper proposes  the Follow-the-Regularized-Leader with a Shannon entropy regularizer, which is built upon the BOBW result of the contextual linear bandit. In addition to regret guarantees, the authors also propose the efficient projection subroutine in any FTRL with a Legendre regularizer. They show the reduction of the typical K-dimensional convex projection to a one-dimensional root-finding problem by KKT conditions. Finally, they provided the empirical experimental evaluations of the proposed methods.

**Strengths:**

- First result of the best-of-both-worlds algorithm for contextual combinatorial bandits.
- The novel choice of the Shannon entropy regularizer, which is defined over the continuous actions of conv(Action space).
- The computational advantage of the projection subroutine can be applicable to any FTRL framework in combinatorial semi-bandits. This advantage of Bisection over MOSEK/Newton is empirically validated.

**Weaknesses:**

- The problem formulation for adversarial combinatorial semi-bandits is based on Zierahn et al. (2023). However, a comparison with Zierahn et al. (2023) is missing in the empirical evaluation.
- There are several formulations in  existing works on Contextual combinatorial bandits. Qin et al. (2014) is another formulation for the stochastic setting while Zierahn et al. (2023) is only for the adversarial setting. The paper needs to discuss the regret bound for stochastic regimes more carefully.
- Due to the use of Shannon entropy regularizer and Precision-matrix estimation, the regret bound in the stochastic regime is $O(\log^3 T)$ as $\kappa=O(\log T)$. Arguments of $O(\log {T} )$ bound in the abstract/Introduction might be overclaimed.
- The algorithm design and theoretical analysis is built upon the contextual linear bandits of Kuroki et al. (2024). The main difference is induced by the fact that Shannon entropy is now defined over the convex hull of the combinatorial action set. Given the fact that the proposed algorithm is not applicable to the case with $\alpha$-approximate oracles for the offline comibinatorial problem, the techniques and theoretical analysis employed here is not significantly different from known results.

**Questions:**

- Does the stochastic regime coincide with the setting by Qin et al. (2014)? Is so, could you provide the comparison with existing regret bounds? If not, could you describe what is the existing bound and also discuss the difference in the existing problem setting including the assumptions. For example, many stochastic combinatorial bandits assume $R$-sub Gaussian noise, but in the paper, you can only deal with the bounded noise due to the technical reason in the best-of-both-worlds algorithms. Is the context space allowed to be infinite?
- Regarding the reduction of K-dimensional projection to a 1D root-finding using KKT, in special structures such as spanning tree or matroid bases, one could instead apply a Frank–Wolfe update to obtain a convex decomposition at extreme points in the FTRL framework.
Frank–Wolfe update is already wildly applicable for various combinatorial actions.  Could you comment on whether the KKT-based projection offers additional benefits or theoretical guarantees beyond such structure-dependent methods?

---

> ### Author Response · Authors · 2025-11-21
>
> Thank you for your thoughtful evaluation of our work. We first address the key weaknesses you identified, and then respond to your specific questions point-by-point.
>
> **Empirical comparison with Zierahn et al. (2023).** Thank you for the suggestion. Below, we present additional experimental results demonstrating that our regret bound outperforms that of Zierahn et al. (2023). The observed performance gain stems from the time-varying learning rate.
>
> **Final cumulative regret (Negative Shannon entropy regularizer)**
>
> | gap  | dim | m-set | Method            | Stochastic $(\mu\pm\sigma)$ | Adversarial $(\mu\pm\sigma)$ |
> |------|-----|--------|-------------------|------------------------------|-------------------------------|
> | 0.125 | 16 | 3 | Bisection (Ours) | $652.3 \pm 110.5$ | $626.2 \pm 89.1$ |
> |       |    |   | Fixed $\eta$     | $708.3 \pm 78.5$  | $767.4 \pm 78.9$ |
> | 0.25  | 16 | 3 | Bisection (Ours) | $464.3 \pm 62.7$  | $499.3 \pm 118.2$ |
> |       |    |   | Fixed $\eta$     | $690.9 \pm 102.0$ | $672.5 \pm 55.9$ |
> | 0.5   | 16 | 3 | Bisection (Ours) | $271.7 \pm 84.9$  | $258.3 \pm 70.5$ |
> |       |    |   | Fixed $\eta$     | $608.9 \pm 84.6$  | $592.8 \pm 55.4$ |
>
> **Regret bound comparison with Qin et al. (2014).** Thank you for raising this point. The algorithm of Qin et al. (2014) is based on an upper-confidence-bound (UCB) framework and, as a result, its regret guarantee holds only in the stochastic regime. Once an adversary can arbitrarily manipulate rewards over time, the confidence intervals become invalid and the analysis fails.
> The sub-Gaussian noise assumption is therefore crucial for their approach, as it ensures valid confidence bounds. In contrast, our regret analysis does not rely on confidence bounds. Instead, we use a self-bounding argument, which only requires that the observed rewards are drawn from a fixed distribution; no reward-boundedness assumption is actually needed, as we verified when revisiting the proofs.
> Qin et al. (2014) assume bounded contexts (see their Theorem 4.2), and we make the same bounded-context assumption. They treat both linear and nonlinear reward models but impose additional structural conditions such as monotonicity and Lipschitz continuity. By contrast, the techniques developed in our work for linear rewards extend naturally to generalized linear models and broader convex reward classes, provided that the underlying parameter can be learned via an FTRL-type method. Moreover, random feature maps can be used to uniformly approximate a potentially nonlinear function class with bounded RKHS norm (Rahimi and Recht, 2007), further widening the applicability of our approach. Finally, the stochastic regret bound of Qin et al. is only $\widetilde{O}(\sqrt{T})$, whereas ours achieves $\widetilde{O}(\log T)$.
> Ali Rahimi and Benjamin Recht. Random features for large-scale kernel machines. NeurIPS 2007.
>
> **Overclaim of the $\boldsymbol{\log T}$ bound in the abstract and introduction.** Thank you for raising this concern. To the best of our knowledge, every logarithmic regret bound stated in the abstract and introduction is presented using the $\widetilde{O}$ notation. This follows the standard convention used in the literature (see the references cited thereafter). Typically, this refers to $\log T$ as well as any form of polynomial $\log T$ being summarized within the $\widetilde{O}$ notation. If there is a specific passage where you believe this convention should be emphasized more clearly, or where a mistake may have occurred, we would be happy to update the notation; please point us to the exact formulation.
>
> - Agrawal, Shipra, and Navin Goyal. Thompson sampling for contextual bandits with linear payoffs. ICML 2013.
> - Zimmert, Julian, and Tor Lattimore. Return of the bias: Almost minimax optimal high probability bounds for adversarial linear bandits. COLT 2022.
> - Zierahn, Lukas, Dirk van der Hoeven, Nicolo Cesa-Bianchi, and Gergely Neu. Nonstochastic contextual combinatorial bandits. AISTATS 2023.

---

> > ### Author Response · Authors · 2025-11-21
> >
> > **Comparison with Kuroki et al. (2024).** The scope and technical challenges of the two settings are notably different when examined in detail. However, we agree that there are important similarities, particularly in the use of the self-bounding technique of Zimmert and Seldin (2019), which was also adopted in their work.
> > Kuroki et al. (2024) consider a special case of our general framework, much like linear contextual bandits are a restricted instance of combinatorial contextual bandits. In particular, their setting does not extend naturally to the more realistic semi-bandit regime. As discussed in the paragraph before Lemma 2.4, a direct application of their results — or more broadly, of any linear contextual bandit result — to the semi-bandit setting is not straightforward. The main obstacle is the entropy bound, which underpins the derivation of the $\ln T$ regret rate in the stochastic regime.
> > To address this gap, we derive a refined entropy bound by (i) carefully partitioning the base arm set $[K]$, and (ii) lifting the mean-action space (support size $K$) to the action-distribution space (support size exponential in $m$). This extension is crucial to enable the self-bounding technique in the combinatorial setting.
> > Beyond the difference in scope, Kuroki et al. (2024) do not address computational efficiency. In contrast, our approach yields an efficient computational scheme that applies not only to Shannon entropy but more broadly to Bregman projections. This provides a reusable subroutine for a family of bandit algorithms. Moreover, as we elaborate in the next point, our method also significantly outperforms generalized conditional gradient methods.
> >
> > **Comparison to Frank-Wolfe.**
> > Thank you for raising this excellent point. Below we compare Algorithm 3 with an efficient Frank-Wolfe method by Garber and Wolf (2021) (termed NEP-FW). Their NEP-FW achieves a convergence rate of $O\left(M \log \left(C_{\perp} / \epsilon\right)\right)$ with $M$ a problem dependent constant and $C$ universal constant, whereas Algorithm 3 achieves $O(\log (C / \epsilon))$ with $C$ being universal constant. More specifically, every iteration of NEP-FW consists of two subroutines. First is a linear minimization oracle running in $O(K)$ time that selects the $m$ optimal coordinates from the $K$-dimensional polytope. Second is a projection step that requires $O\left(m^3 \log ^3 K\right)$ operations to solve the constrained quadratic program over the convex hull of active vertices, where the number of active vertices is provably bounded by $O(m \log K)$. Moreover, its practical implementation requires careful consideration of the problem-dependent constant $M$, which can dominate convergence in finite-precision applications. Therefore, both the iteration complexity and arithmetic operations required within each iteration of NEP-FW are worse than ours assuming loss vectors are fixed. In our initial experiments, NEP-FW performed poorly in terms of regret under our parameter setup, requiring further tuning of the learning rate to achieve the regret performance reported in our work, and it also exhibited slower run times compared to our bisection algorithm. However, we acknowledge that for certain matroid structures and with carefully chosen hyperparameters, a Frank–Wolfe–type method could outperform our approach in terms of runtime in practice (though not necessarily in terms of theoretical guarantees).
> > - Dan Garber and Noam Wolf. Frank-Wolfe with a Nearest Extreme Point Oracle. COLT 2021.

---

> > > ### Comment · Reviewer_vDJH · 2025-11-27
> > >
> > > Thank you for the response.
> > >
> > >
> > > I appreciate the comparison with Zierahn et al. The improvement is surprisingly better. Please also describe the detailed settings for exploration parameters and learning rates in the revised version.
> > >
> > >
> > > Comment: Regarding Table 1, it should be limited to existing work with the same setting. If Ito et al. (2022) and Kong et al. (2023) are not applicable to contextual combinatorial semi-bandits, it would be better to remove them from the table (discussing them in the main text is fine).
> > >
> > >
> > > Comment:  As Reviewer PYJa also pointed out, the notation is misleading. For the stochastic regime, the main term would be written as $O(\ln^3 T)$. Since $\tilde{O}$ is used to ignore log terms, writing $\tilde{O}(\ln T)$ is inappropriate here. For the adversarial regime, $\tilde{O}$ is acceptable since $\sqrt{T}$ is the main term.
> > >
> > >
> > > Thank you for the discussion on Frank-Wolfe. I still have concerns about significance as the proposed method seems effective only for size $m$-sets.
> > >
> > >
> > > Since several concerns were resolved, I raised my score by 1.

---

> > > > ### Author Response · Authors · 2025-11-28
> > > >
> > > > Thank you very much for taking the time to review our comments and updated manuscript.
> > > >
> > > > For the simulation, we used a fixed learning rate $\eta = 1/\sqrt{T}$ and $\eta_t = 1/\sqrt{t}$ for the time-varying case. In addition, we set the simulation horizon to $T = 10{,}000$. The performance improvement can be attributed to the fact that the fully adversarial setting is hard to simulate. As in other works, we therefore employed the adversarially constrained stochastic setting, as in Zimmert and Seldin (2019). We will include these details in the final version.
> > > >
> > > > Regarding the updated table, we are happy to adjust it accordingly in the final version.
> > > >
> > > > Regarding the notation, we will change $\tilde{O}(\ln T)$ to $\mathcal{O}(\mathrm{polylog}(T))$ to make this clearer.
> > > >
> > > > Regarding your concern about limited significance about the computational efficiency (our second contribution) compared to Frank-Wolfe Algorithm, and given your feedback, we have extended the results to the largest possible class of matroids allowed by our approach. Our efficient Bregman projection method is effective for both uniform and partition matroids, as discussed in Section 3, which already extends beyond the $m$-set setting. More broadly, any matroid whose base constraints can be encoded via a single dual variable per block admits the speedup. Examples include uniform matroids (all base arms sharing one dual variable), partition matroids (one dual per part), laminar matroids (a hierarchy of nested duals), and graphic matroids (dual variables per connected component in a spanning-tree constraint). When the action set does admit a separable structure, bisection is much more efficient than, e.g., Frank–Wolfe: one solves a scalar root-finding problem per dual variable instead of running a costly conditional gradient over the entire polytope. Ultimately, choosing between Frank–Wolfe and bisection is a design choice.

---

### Official Review · Reviewer_4Chh · 2025-10-28

**Soundness:** 3
**Presentation:** 3
**Contribution:** 2
**Rating:** 4
**Confidence:** 4

**Summary:**

This work studies the best-of-both-worlds (BOBW) algorithm for linear contextual combinatorial bandits. The algorithm leverages the follow-the-regularized-leader (FTRL) framework with a data-dependent learning rate to achieve the BOBW guarantee. Due to the monotonicity of the first-order derivative, the authors also prove that the optimization problem of FTRL can be efficiently approximately solved using KKT conditions and bisection. Experiments on the running time and cumulative regret of the proposed algorithm are conducted.

**Strengths:**

1. This work seems to be the first work in the literature to study the BOBW linear contextual combinatorial bandit.
2. Most parts of this work are generally well-written.

**Weaknesses:**

1. **Motivation**: I understand that the linear contextual combinatorial problem might be of some practical interest. However, given (a large number of) previous advances for BOBW combinatorial bandits, linear bandits, and linear contextual bandits, at this point, studying the BOBW linear contextual combinatorial problem seems somewhat not so technically appealing.
2. **Novelty**: My main concern is about the novelty of this work, which is also partially related to the concern above. In recent years, it has been widely observed in various bandit problems that data-dependent learning rates can lead to BOBW regret guarantees. For this work, though the problem formulation is new, the used data-dependent learning rate $\beta_t$ has a very similar form to those in previous works studying BOBW graph bandit [1], linear bandit [2], and linear contextual bandit [3]. The resulting intermediate regret bound (Lemma 2.4), depending on the summation of the entropy values, is also very similar to those in previous works. In this work, the use of FTRL with Shannon entropy over the convex set of super-arms, instead of the space of distributions over super-arms, has already been proposed in previous work for the adversarial contextual combinatorial problem [4]. Also, the used techniques for contextual bandits, including matrix geometric resampling and ghost context sampling, are standard in the literature for contextual bandits [5]. Therefore, to me, the main contribution of this work is that the authors find that the optimization problem of FTRL with Shannon entropy over the convex set of super-arms in special cases of $m$-set and partition matroid can be efficiently addressed by leveraging KKT condition and the monotonicity of the first-order derivative of the regularizer (btw, the definition of $m$-set seems to be that $\sum\_{k=1}^K(A)\_k = m $ instead of $ \sum\_{k=1}^K(A)\_k \leq m$).
Therefore, from the viewpoint of bandit learning, the novelty of the proposed algorithm and results appears to be limited. On the other hand, I am not working in the optimization area, so I think my sense for evaluating the novelty of the approximate solution to the FTRL optimization considered in this work might not be accurate. However, I also personally feel that using the KKT condition to solve a constrained optimization is standard, which I believe appears in most (basic) convex optimization courses.
3. **Presentation**: It is good to see there are many discussions about the computation details of the FTRL update. However, for bandit learning, I personally think that the discussions and comparisons between regret bounds might be the most important parts to be included in the main paper body. For example, when specializing the results in the special cases of linear contextual combinatorial bandits, what are the advantages and disadvantages of the regret bound in this work when compared with previous works studying BOBW combinatorial bandits [6], linear bandits [2], and linear contextual bandits [3]?

[1] Ito et al. Nearly Optimal Best-of-Both-Worlds Algorithms for Online Learning with Feedback Graphs. NeurIPS, 22.

[2] Kong et al. Best-of-three-worlds Analysis for Linear Bandits with Follow-the-regularized-leader Algorithm. COLT, 23.

[3] Kuroki et al. Best-of-Both-Worlds Algorithms for Linear Contextual Bandits. AISTATS, 24.

[4] Zierahn et al. Nonstochastic Contextual Combinatorial Bandits. AISTATS, 23.

[5] Neu et al. Efficient and robust algorithms for adversarial linear contextual bandit. COLT, 20.

[6] Zimmert et al. Beating Stochastic and Adversarial Semi-bandits Optimally and Simultaneously. ICML, 19.

**Questions:**

1. Typically, FTRL with Shannon entropy regularization requires updating the policy distribution over the action space. This is computationally inefficient for linear bandits if action sets are exponentially large. To avoid this drawback, previous works have leveraged FTRL with self-concordant regularizers, which is operated in the convex set of the action feature set and is even shown to be possible to achieve the BOBW bound for linear bandits [7]. Is FTRL with a self-concordant regularizer also applicable to the linear contextual combinatorial problem?
2. I notice that the dependence on $\log T$ of the proposed algorithm in this work seems to be slightly better than previous works using FTRL with Shannon entropy regularizer (say, [1,2]). Could the authors comment more on this?

[7] Ito et al. Best-of-Three-Worlds Linear Bandit Algorithm with Variance-Adaptive Regret Bounds. COLT, 23.

---

> ### Author Response · Authors · 2025-11-21
>
> Thank you for your detailed feedback.
>
> **Motivation for problem selection.** We believe that, beyond the practical relevance outlined in the introduction, there is also clear theoretical interest in achieving optimal regret bounds for such a general and expressive problem class. Establishing optimal guarantees in this most challenging setting typically requires new proof techniques that can then inform and strengthen results for important subclasses (we elaborate on this in our response on “Novelty,” where we explain why existing techniques do not directly apply).
> Moreover, contextual bandits remain highly relevant to ICLR, as evidenced by the community’s sustained interest. As a strict subset of reinforcement learning with action-independent state transitions, bandits capture a key class of problems outside the standard Markovian framework. They also form the theoretical and algorithmic foundation for RL, providing the core machinery for exploration under uncertainty. Our best-of-both-worlds results for stochastic and adversarial regimes contribute directly to this line of work.
>
> **Novelty.** We stress that we are interested in solving one of the most general cases in terms of regret bounds, addressing stochastic, and adversarial in the face of context and combinatorial action sets. Naturally, the proof uses techniques for these subclasses. However, applying the framework deployed by Kuroki et al. (2024) to our setting (so as any linear contextual bandit regret bounds to contextual semi-bandit regret bounds) is not straightforward. At its crux is the entropy bound that empowers the $\ln T$ regret bound in the stochastic regime. To overcome these limitations, we derive a refined entropy bound by exploiting a careful partitioning of the base arm set $[K]$ and lifting the mean-action space (support size $K$) to the space of action distributions (support size exponential in $m$).
> Aside from these technical hurdles and differences in terms of scope, the computational efficiency is not addressed at all by Kuroki et al. (2024). Our computational scheme not only works for the Shannon entropy but in general provides a new approach to solving the Bregman projection step that is useful as a subroutine to a family of bandit problems. In our next point, we also point out that our method beats a generalized conditional gradient method with a large margin.
>
> **Table of regret bound comparison.** Thank you for the suggestion. We have added a comparison table at the end of Section 1.1 and also include it below for convenience, which we hope provides additional clarity regarding our contributions.
>
> |                     | Our paper                                              | Qin et al. (2014)                         | Zierahn et al. (2023)                                       | Ito et al. (2022)                            | Kong et al. (2023)                          |
> |---------------------|--------------------------------------------------------|-------------------------------------------|-------------------------------------------------------------|----------------------------------------------|---------------------------------------------|
> | **Feedback**        | Semi-bandit                                           | Full-bandit                               | Semi/Full-bandit                                            | Graph bandit                                 | Linear bandit                               |
> | **Adv. regret**     | $\widetilde{\mathcal{O}}\left(\mathrm{poly}(d,m,K)\sqrt{T}\right)$ | N/A                                       | $\widetilde{\mathcal{O}}\left(\mathrm{poly}(d,m,K)\sqrt{T}\right)$ | $\widetilde{\mathcal{O}}\left(\sqrt{\alpha T}\right)$ | $\widetilde{\mathcal{O}}\left(\sqrt{T}\right)$ |
> | **Stoch. regret**   | $\mathcal{O}\left(\mathrm{poly}(d,m,K)(\ln T)^3\right)$         | $\widetilde{\mathcal{O}}\left(d\sqrt{mT}\right)$ | N/A                                                         | $\mathcal{O}\left(\dfrac{\alpha(\ln T)^3}{\Delta_{\min}}\right)$ | $\mathcal{O}\left(\dfrac{(\ln T)^2}{\Delta_{\min}}\right)$ |

---

> > ### Author Response · Authors · 2025-11-21
> >
> > **FTRL with self-concordant regularizers.** Thank you very much for raising this excellent point. Indeed, sampling from a generic combinatorial action set may be computationally demanding and FTRL with self-concordant regularizers would help because sampling from the policy distribution is equivalent to sampling from a Dikin ellipsoid. However, it is not always easy to find a corresponding self-concordant barrier for an arbitrary convex set in the first place. In addition, the Hessian structure required for self-concordance is no longer preserved when projecting back from the mean-action polytope to a distribution over the entire action space, and the essential entropy bound in Lemma 2.4 is not straightforward to extend in this case. Due to these two reasons, we don’t see a clear path generalizing the idea of self-concordant barrier from linear to combinatorial bandits, but we would be more than happy to add a related discussion in the conclusion.
> > We stress that we do not need to explicitly compute the policy distribution for separable matroids. In particular, we employ an efficient sampling oracle of complexity $\widetilde{\mathcal O}(K)$ proposed by (Zimmert and Seldin 2019) in the case of $m$-sets, the efficiency of our proposed FTRL algorithm is comparable to the sampling procedure employed by FTPL (perceived to be the most computationally efficient approach to semi-bandits) for the same class of problems (Lattimore and Szepesvari 2020).
> >
> > **$\boldsymbol{\log T}$ dependence.** Thank you for raising your concern and providing the references [1,2]. In terms of $\log T$ dependence, we believe we have one extra $\log T$ term compared to [2, Theorem 1], and we have the same $\log T$ dependence compared to [1, Theorem 1].

---

### Official Review · Reviewer_PYJa · 2025-10-31

**Soundness:** 3
**Presentation:** 1
**Contribution:** 2
**Rating:** 4
**Confidence:** 4

**Summary:**

This paper studies a special case of contextual combinatorial semi-bandits and provides best-of-both-world guarantees for it. The setting is as follows: At each time step $t$, a $d$-dimensional context $X_t$ is revealed, then the learner has to select $m$ out of $K$ actions. Each action $k$ is associated with a $d$-dimensional vector $\theta_{t,k}$, whose scalar product with $X_t$ defines its loss at time $t$. The instantaneous loss incurred by the learner is the sum of the losses of the action chosen at that time step. The total loss of the learner is given by the sum of the instantaneous losses, and it is compared to the total loss of the optimal context-dependent action map that achieves minimum loss in hindsight.

The authors study the (oblivious) adversarial setting, where the $\theta_{t,k}$ vectors are chosen up-front by an adversary in an arbitrary way, the stochastic setting, where the  $\theta_{t,k}$ are drawn i.i.d. from a fixed distribution for each action, and the corrupted case, where such distributions may slowly change over time, and some unbiased noise is added at each iteration.

The main contribution of the paper is a best-of-both-worlds algorithm, which the authors claim has an $O(\sqrt{T})$ regret bound in the stochastic setting, and an $O(\log T)$ instance-dependent bound otherwise.

**Strengths:**

- The authors complemented their theoretical results with some experiments. Although this is not strictly mandatory for theoretical online learning submissions, it is a nice addition.
- To the best of my understanding, the paper is correct
- Best-of-both-world results are important and of practical value
- The authors also propose a numerical speed-up that enhances applicability

**Weaknesses:**

- The topic of the paper is extremely narrow, as the problem studied is quite involved and specialized. I am unsure whether it will garner general interest among the vast ICLR audience.
- It is very challenging to gain a comprehensive understanding of the state-of-the-art and to compare it with the results presented in the paper. There are many parameters (namely, the context dimension $d$, the number of base actions $K$, and the cardinality constraint $m$), as well as numerous results implied by past works. Therefore, I strongly suggest that the authors include a table comparing the regret rates across various settings/assumptions.
- The modeling assumption that the contexts are independent and identically distributed (i.i.d.) is somewhat surprising. What can be said when they are generated adversarially as well?
- The overall algorithm construction is non-trivial but not novel, as it follows the standard FTRL template for BOBW.
- My main issue is with the presentation of the results, which seems to me quite misleading: in the statement of Theorem 1, there is a parameter $\kappa$ that inexplicably hides some important terms, making the understanding of the regret rates complicated.  $\kappa \in \Theta(\log T)$ term, this means that the stochastic regret is $O(\log^2T)$, and not $\log T$ as claimed in the introduction. The term $\kappa$ also depends linearly on $\sqrt{K}$, so that the overall regret rates depend polynomially on $K$, and not only on $m$, which may be drastically smaller.

Minor:
- The title ``Combinatorial Semi-Bandits'' is a bit misleading, as I was expecting a model similar to the one in the seminal paper by Cesa-Bianchi and Lugosi, where the action set could have any combinatorial structure (thus not only the cardinality constraint proposed in this paper). I am aware that other papers have already employed the generic combinatorial term for this specific case, but I still find it somewhat unusual.
- In line 99, the authors refer to a lower bound by Bubeck & Cesa-Bianchi (2012), but I think that the citation is to the wrong paper. Shouldn’t it be Bubeck, Cesa-Bianchi, and Kakade COLT 12? In particular, does the cited lower bound apply to the special case of combinatorial bandits studied in this paper?

**Questions:**

Please address my comments above. In particular, am I missing something about the $\kappa$ parameter?

---

> ### Author Response · Authors · 2025-11-21
>
> We are grateful for your insights. We begin by addressing your primary concerns and then answer your questions individually.
>
> **Topic of the paper.** We respectfully disagree with the comment that the problem studied is involved and specialized. We believe contextual bandits are highly relevant to ICLR, as evidenced by the community's ongoing engagement. As a strict subset of reinforcement learning where state transitions are action-independent, bandits address a key class of problems outside the standard Markovian, sequential framework. Bandits form the theoretical and algorithmic foundation for RL, providing the core machinery for exploration under uncertainty. Our research on best-of-both-worlds algorithms for stochastic and adversarial settings contributes directly to this area. The significant interest in this subfield is confirmed by its presence at ICLR 2025, which featured 21 papers on bandit problems, 7 of which focus specifically on contextual bandits. This underscores the topic's clear alignment with the conference's scope.
>
> **Table of comparison with most relevant related works.** Thank you for your comment. We have added a comparison table with the main related works at the end of Section 1.1, summarizing the bounds for stochastic and adversarial bandit settings and the types of feedback considered in each model.
>
> |                     | Our paper                                              | Qin et al. (2014)                         | Zierahn et al. (2023)                                       | Ito et al. (2022)                            | Kong et al. (2023)                          |
> |---------------------|--------------------------------------------------------|-------------------------------------------|-------------------------------------------------------------|----------------------------------------------|---------------------------------------------|
> | **Feedback**        | Semi-bandit                                           | Full-bandit                               | Semi/Full-bandit                                            | Graph bandit                                 | Linear bandit                               |
> | **Adv. regret**     | $\widetilde{\mathcal{O}}\left(\mathrm{poly}(d,m,K)\sqrt{T}\right)$ | N/A                                       | $\widetilde{\mathcal{O}}\left(\mathrm{poly}(d,m,K)\sqrt{T}\right)$ | $\widetilde{\mathcal{O}}\left(\sqrt{\alpha T}\right)$ | $\widetilde{\mathcal{O}}\left(\sqrt{T}\right)$ |
> | **Stoch. regret**   | $\mathcal{O}\left(\mathrm{poly}(d,m,K)(\ln T)^3\right)$         | $\widetilde{\mathcal{O}}\left(d\sqrt{mT}\right)$ | N/A                                                         | $\mathcal{O}\left(\dfrac{\alpha(\ln T)^3}{\Delta_{\min}}\right)$ | $\mathcal{O}\left(\dfrac{(\ln T)^2}{\Delta_{\min}}\right)$ |
>
>
> **i.i.d. Contexts.** Our results cover the three settings, namely i.i.d. context with adversarial, corrupted stochastic or stochastic linear rewards. We assume you are referring to the case of adversarial contexts with adversarial linear rewards. In this case, Neu and Olkhovskaya (2020) argued that the problem is at least as hard as online learning a one-dimensional threshold function, for which sublinear regret is impossible. Hence, due to this limitation, the literature typically follows the i.i.d. context assumption, as we do in our work.
> - Gergely Neu and Julia Olkhovskaya. Efficient and robust algorithms for adversarial linear contextual bandits. COLT 2020.
>
> **FTRL framework for BOBW.** We agree that the algorithm design falls into the same FTRL framework as the majority of the online learning works. However, we believe this does not constitute the ground for rejection. We argue that the result for the more generalized problem of combinatorial semi-bandit is new, and from the technical side, the entropy bound presented in Lemma 2.4 is new as well as the proposed efficient computational subroutine.
>
> **“combinatorial semi-bandits”.**
> Our regret bound generalizes naturally to the full combinatorial semi-bandits setting, where we simply set $m\coloneqq\max_{A\in\mathcal{A}} \|A\|_1$.
>
> We have updated our paper to reflect the general action set structure $\mathcal{A} \subseteq\{A\in \lbrace 0,1 \rbrace^K : \sum_{k=1}^K (A)_k\le m\}$ and added a clarification in the contribution part.
>
> **Citation Bubeck & Cesa-Bianchi (2012).**
> Thank you for raising this point. We have carefully reviewed the citation, and our assessment is that it is correct in its current form. We have also added a more precise pointer to the relevant section, as well as the original citation Audibert et al. (2014), which is also referenced in Bubeck and Cesa-Bianchi (2012). If we have misunderstood your concern, we would be grateful for further clarification.

---

### Official Review · Reviewer_SW5m · 2025-11-07

**Soundness:** 3
**Presentation:** 3
**Contribution:** 2
**Rating:** 6
**Confidence:** 3

**Summary:**

The paper proposes an FTRL-based algorithm for contextual combinatorial semi-bandits. It achieves best-of-both-worlds (BOBW) regret: $O(\sqrt{T})$in adversarial settings and $O(\log T)$ in corrupted stochastic settings.
It uses a Shannon-entropy regularizer together with an efficient projection routine derived from the KKT conditions over the convex hull of the m-set. This reduction converts a high-dimensional optimization into a one-dimensional root-finding task per round, yielding computational savings. Empirical evaluations demonstrate the effectiveness of the proposed method.

**Strengths:**

1) This paper presents the first best-of-both-worlds (BOBW) regret guarantee for contextual combinatorial semi-bandits.
2) It introduces an efficient numerical scheme for the FTRL update that significantly reduces computational cost while preserving the theoretical guarantees.
3) The theoretical analysis is well-structured, with a new auxiliary “ghost context” game that simplifies the regret analysis.
4) Empirical evaluations demonstrate substantial runtime improvements compared with Newton and MOSEK solvers.

**Weaknesses:**

1) This paper assumes a linear reward (or loss) function. The framework does not yet handle nonlinear or generalized linear models.
2) The proposed efficient projection scheme is tailored to the m-set (and potentially partition matroids). For more general combinatorial structures, this computational savings may no longer hold.
3) The Shannon-entropy regularizer adds an additional $O(\log T)$ term in the adversarial regret, slightly weakening the asymptotic bound.

**Questions:**

1) The analysis relies on linear reward assumptions. Could the proposed method be extended to generalized linear or nonlinear reward functions (e.g., incorporating a link function)?
2) The efficient projection routine appears tailored to m-sets (and perhaps partition matroids). What are the main challenges for generalization, e.g., non-separability, lack of monotonicity, or absence of closed-form inverses?
3) Shannon entropy has been widely used in FTRL. Is the contribution mainly its specific integration into the BOBW setting or the projection simplification that follows from its analytical form? How does this choice compare with other regularizers employed in similar contexts?

---

> ### Author Response · Authors · 2025-11-21
>
> Thank you for your helpful comments.
>
> **Generalization to generalized linear models.**  Excellent point. We did not explicitly mention the broader generalizability of our theoretical results. Indeed, the techniques developed in this paper for linear reward models extend naturally to generalized linear models and convex reward structures, as long as the underlying parameter can be learned via an FTRL-like method.
> Note that by using random feature mappings, one can uniformly approximate a (potentially nonlinear) function class with bounded norms in a reproducing kernel Hilbert space (Rahimi and Recht 2007).
> We stress that the only relevant works on semi-bandit with nonlinear loss functions are those of Kale et al. (2010) and Krishnamurthy et al. (2016). Both require a finite policy class that maps contexts to combinatorial actions, and otherwise, implementing these algorithms requires either a full enumeration of the exponentially-sized policy space or access to a non-standard optimization oracle. These works highlight the difficulty for addressing the nonlinear reward setting.
>
> - Ali Rahimi and Benjamin Recht. Random features for large-scale kernel machines. NeurIPS 2007.
> - Kale, Satyen, Lev Reyzin, and Robert E. Schapire. Non-stochastic bandit slate problems. NeurIPS 2010.
> - Krishnamurthy, Akshay, Alekh Agarwal, and Miro Dudik. Contextual semibandits via supervised learning oracles. NeurIPS 2016.
>
> **Main challenges for generalization to other matroid classes.** Thank you for raising this point. Indeed our computational subroutine is specialized to the case where a separable objective is present in the Bregman objective.
> - **Non-separability** leads to dual/KKT conditions that no longer reduce to a one-dimensional root-finding problem; instead, the full problem dimension must be retained.
> - **Loss of monotonicity in the derivative** introduces a non-convex regularizer in the Bregman projection, making the problem NP-hard.
> - **Absence of closed-form inverses** can still be handled: we can employ an approximation oracle for inverting the relevant operators. This preserves theoretical computational efficiency, and our experiments indicate that similar practical speed-ups are achievable.
>
> **Selection of Shannon entropy.** The choice of regularizer must strike a balance between computational tractability and statistical efficiency (i.e., the resulting regret bound). Our use of Shannon entropy provides a favorable trade-off: it enables efficient computation while only incurring a $\log(T)$ regret rate.
> By contrast, Tsallis entropy is often adopted to achieve the theoretically optimal regret bound. However, its sharp curvature near zero typically leads to less stable and slower optimization in practice. This makes the computational procedure substantially more difficult, even though the statistical guarantee is stronger.
> For these reasons, we believe Shannon entropy offers a well-balanced choice — it preserves statistical guarantees of practical relevance while enabling a computationally efficient learning procedure.

---

### Author Response · Authors · 2025-12-02

We would like to express our genuine gratitude to the Area Chair for handling our submission and to the reviewers for their careful reading and thoughtful feedback. To make our position easy to assess, we summarize below the main strengths identified, the key concerns raised, and how our rebuttal addresses them.

Across the reviews, the core strengths are consistently acknowledged.
- Reviewers SW5m, 4Chh, vDJH recognize that this is the **first BOBW result for contextual combinatorial semi-bandits.**
- Reviewers PYJa, vDJH, SW5m highlight that the **computational contribution yields substantial per-round speed-ups.**

The main concerns are:

**(1) Relation to prior BOBW work (Reviewers PYJa, 4Chh, vDJH).**
We addressed this by:
- Clarifying our positioning: contextual combinatorial semi-bandits form a very general bandit framework, subsuming many standard models (e.g., contextual linear, combinatorial, and semi-bandits). Within this general setting, we explicitly note that our regret bounds cover stochastic, corrupted-stochastic, and adversarial regimes under semi-bandit feedback with **general combinatorial action sets**. We see our contribution as providing a unified analysis across these uncertainty regimes for generalized action sets, while preserving computational efficiency, thereby complementing existing, more specialized results.
- Explaining that, while we use FTRL, a **key technical contribution** is a new entropy bound (Lemma 2.4) tailored to the combinatorial semi-bandit setting. This requires (i) a careful partitioning of the base-arm set and (ii) lifting from the mean-action space (support size $K$) to the exponential-size distribution space over super-arms, so that the self-bounding technique can be applied in the semi-bandit regime. Existing analyses for linear contextual bandits (e.g., Kuroki et al., 2024) do not directly extend to this setting, precisely because their entropy control is formulated at the level of linear actions and cannot accommodate the combinatorial semi-bandit structure.
- Emphasizing that we also address a **computational gap** not covered in prior BOBW work: we provide an efficient projection routine that applies broadly within FTRL-based combinatorial semi-bandits, and not just to our specific algorithm.
- Adding a table (end of Section 1.1) summarizing the regret bounds and assumptions across our paper and the most relevant baselines.

**2) Scope of the computational subroutine (Reviewers SW5m, 4Chh, vDJH).**
Reviewers asked how far our bisection scheme extends beyond $m$-sets, and how it compares to Frank–Wolfe.
We clarified and strengthened this part by:
- Explaining that our efficient Bregman projection applies to **a broader class of matroids** wherever base constraints can be encoded via a small number of dual variables per block. This includes **uniform matroids, partition matroids**, and extends to more structured cases such as **laminar** and certain **graphic matroids**. We explicitly updated the text to reflect this broader applicability, rather than only to $m$-sets.
- Providing a **detailed theoretical comparison with SOTA Frank–Wolfe–type methods**. We noted that each FW iteration requires both a linear minimization oracle over the combinatorial polytope and a projection over the convex hull of active vertices (whose number grows with iterations), which is more expensive in our setting than solving a scalar root-finding problem.

**(3) Linear rewards, the i.i.d. context and noise assumptions (Reviewers SW5m, PYJa, vDJH).**
We addressed this by:
- Clarifying that the **proof techniques extend naturally** beyond linear rewards to generalized linear models and more general convex reward structures, as long as the underlying parameter can be learned via an FTRL-type method. We also pointed out that **random feature maps** allow us to approximate nonlinear function classes with bounded RKHS norms uniformly.
- Justifying the **i.i.d. context assumption** by referencing a prior result that adversarial contexts with adversarial rewards must suffer linear regret.
- Revisiting our proofs to clarify that we only require rewards to be drawn from fixed distributions, and that we do not rely on strong sub-Gaussian noise assumptions as in some prior work.

In summary, the reviewers agree that the paper is technically sound and addresses a relevant setting. The initial concerns focused on scope, novelty, and presentation of the regret bounds and computational contribution. Through our rebuttal, we:

(i) clarified the position of our work within the broader BOBW literature and emphasized the genuinely new aspects of our entropy-based analysis in the semi-bandit setting;

(ii) made the regret notation and parameter dependence explicit and added a comparison table; and

(iii) demonstrated that our computational subroutine applies to a broad class of combinatorial structures and offers clear benefits over existing alternatives.

---

### Meta-Review · Area_Chair_Rz2a · 2026-01-06

**Summary:**

The paper studies contextual combinatorial semi-bandits in the best-of-both-world setting. The considered combinatorial structure is m-sets, where the agent chooses a set of actions of size (at most) $m$ among $K$ actions. Its main contribution is an FTRL-based algorithm with sound regret guarantees simultaneously in both adversarial (with a regret of $\sqrt{T}$) and (corrupted) stochastic environments (with a polylog($T$) regret). A key aspect in the algorithmic development is use of Shannon entropy in the regularizer, instead of Tsallis entropy, which yields computational efficiency in the considered combinatorial setting, at the expense of an extra logarithmic term in the regret bound.

The reviewers agree that the paper is the first to present and analyze a best-of-both-world algorithm for the considered contextual combinatorial setting. While admiring its computational efficiency (both theoretically and empirically), the reviewers raised some concerns regarding technical novelty in the regret analysis (as the paper builds largely on existing technical tools) as well as limitations to specific combinatorial structure. The rebuttal adequately addressed the most critical concerns raised by the reviewers, and elaborated on positioning the method with respect to existing methods. I therefore recommend acceptance.

**Reviewer Concerns:**

__Use of Shannon entropy instead of Tsallis entropy and its implications.__ This was adequately addressed in the rebuttal. In particular, the related computational-sample efficiency tradeoff was elaborated in the rebuttal.

__Comparing to prior work on BOBW in terms of technical novelty.__ Some reviewers mentioned that techniques for developing and analyzing BOBW algorithms are well-established and by now standard, which limits the technical novelty. The rebuttal adequately addressed this point by highlighting the technical challenges of ensuring computational efficiency in the considered contextual setting.

__Limitations to simple combinatorial settings such as $m$-sets (and related matroids).__ This concern still stands; the rebuttal confirms that the proposed method cannot be extended beyond some matroid structures (e.g., $m$-sets or when choosing at most $m$ actions).

__Relevance to ICLR.__ It was raised in the reviews. I do believe that the topic attracts audience in ICLR.

**Reviewer Scores:**

- Reviewer SW5m: Although the reviewer's concerns are discussed, some concerns still stand. I believe the reviewer would maintain the current score.
- Reviewer PYJa: The reviewer raised some technical and presentational concerns. The rebuttal did a great job to address them, so I think it is likely that the reviewer would increase the score.
- Reviewer 4Chh: Difficult to say. Yet, with a good chance the reviewer would be willing to engage further in the discussion, which would likely lead to increase in the score.
- Reviewer vDJH: The reviewer mentioned in the discussion to have increased their score to the next level --more precisely to 6--, which makes perfect sense considering the rebuttal.

---

### Decision · Program_Chairs · 2026-01-26

Accept (Poster)